# Active clustering for labeling training data

**Quentin Lutz***
Nokia Bell Labs
quentin.lutz@nokia-bell-labs.com

**Élie de Panafieu**
Nokia Bell Labs
elie.de_panafieu@nokia-bell-labs.com

**Alex Scott**
University of Oxford
scott@maths.ox.ac.uk

**Maya Stein**
University of Chile
mstein@dim.uchile.cl

## Abstract

Gathering training data is a key step of any supervised learning task, and it is both critical and expensive. Critical, because the quantity and quality of the training data has a high impact on the performance of the learned function. Expensive, because most practical cases rely on humans-in-the-loop to label the data. The process of determining the correct labels is much more expensive than comparing two items to see whether they belong to the same class. Thus motivated, we propose a setting for training data gathering where the human experts perform the comparatively cheap task of answering pairwise queries, and the computer groups the items into classes (which can be labeled cheaply at the very end of the process). Given the items, we consider two random models for the classes: one where the set partition they form is drawn uniformly, the other one where each item chooses its class independently following a fixed distribution. In the first model, we characterize the algorithms that minimize the average number of queries required to cluster the items and analyze their complexity. In the second model, we analyze a specific algorithm family, propose as a conjecture that they reach the minimum average number of queries and compare their performance to a random approach. We also propose solutions to handle errors or inconsistencies in the experts' answers.

## 1 Introduction

There is an increasing demand for software implementing supervised learning for classification. Training data input for such software consists of items belonging to distinct classes. The output is a classifier: a function that predicts, for any new item, the class it most likely belongs to. Its quality depends critically on the available learning data, in terms of both quantity and quality [21]. But labeling large quantities of data is costly. This task cannot be fully automated, as doing so would assume access to an already trained classifier. Thus, human intervention, although expensive, is required. In this article, we focus on helping the human experts build the learning data efficiently.

One natural way for the human experts to proceed is to learn (or discover) the classes and write down their characteristics. Then, items are considered one by one, assigning an existing class to each of them, or creating a new one if necessary. This approach requires the experts to learn the various classes, which, depending on the use-case, can be difficult. A different approach, proposed to us by Nokia engineer Maria Laura Maag, is to discover the partition by querying the experts on two items at a time asking whether these belong to the same class or not. This approach avoids the part of the process where classes are learned, and can therefore be cheaper. It is the setting we consider here, and we call the corresponding algorithm an *active clustering algorithm*, or for short, *AC algorithm*.

---

*Authors presented in alphabetical order.

35th Conference on Neural Information Processing Systems (NeurIPS 2021).

More precisely, we assume there is a set of size $n$, with a partition unknown to us. An *AC algorithm* will, in each step, choose a pair of elements and asks the oracle whether they belong to the same partition class or not. The choices of the queries of the algorithm are allowed to depend on earlier answers. The algorithm will use transitivity inside the partition classes: if each of the pairs $x$, $y$ and $y$, $z$ is known to lie in the same class (for instance because of positive answers from the oracle), then the algorithm will 'know' that $x$ and $z$ are also in the same class, and it will not submit a query for this pair. The algorithm terminates once the partition is recovered, i.e. when all elements from the same partition class have been shown to belong to the same class, and when for each pair of partition classes, there has been at least one query between their elements.

We investigate AC algorithms under two different random models for the unknown set partition. In Section 2.1, the set partition is sampled uniformly at random, while in Section 2.2, the number of blocks is fixed and each item chooses its class independently following the same distribution. Section 2.3 analyzes the cases where the experts' answers contain errors or inconsistencies. Our proofs, sketched in Section 3, rely on a broad variety of mathematical tools: probability theory, graph theory and analytic combinatorics. We conclude in Section 4, with some interesting open problems and research topics.

**Related works.** Note that our model has similarities with sorting algorithms. Instead of an array of elements equipped with an order, we consider a set of items equipped with a set partition structure. In the sorting algorithm, the oracle determines the order of a pair of elements, while in our setting, the oracle tells whether two items belong to the same class. Both in sorting and AC algorithms, the goal is the design of algorithms using few queries.

The cost of annotating raw data to turn it into training data motivated the exploration of several variants of supervised learning. *Transfer learning* reduces the quantity of labeled data required by using the knowledge gained while solving a different but related problem. *Semi-supervised learning* reduces it by learning also from the inherent clustering present in unlabeled data. In *active learning*, the learner chooses the data needing labeling and the goal is to maximize the learning potential for a given small number of queries. However, users who want to use supervised learning software for classification as a black-box can mitigate the annotating cost only by modifying the labeling process.

Recent papers [9, 18] acknowledge that humans prefer pairwise queries over pointwise queries as they are better suited for comparisons. Pairwise queries have been considered in semi-supervised clustering [5, 24, 16] where they are called *must-link* and *cannot-link* constraints. The pairs of vertices linked by those constraints are random in general, but chosen adaptively in active semi-supervised clustering [4]. In both cases, the existence of a similarity measure between items is assumed, and a general theoretical study of this setting is provided by [19]. This is not the case in [11, 17], where a hierarchical clustering is built by successively choosing pairs of items and measuring their similarity. There, the trade-off between the number of measurements and the accuracy of the hierarchical clustering is investigated. The difference with the current paper is that the similarity measure takes real values, while we consider Boolean queries, and that their output is a hierarchical clustering, while our clustering (a set partition) is flat.

The problem we consider is motivated by its application to annotation for supervised machine learning. It also belongs to the field of *combinatorial search* [1, 7]. Related papers [2, 15] consider the problem of reconstructing a set partition using queries on sets of elements, where the answer to such a query is whether there is an edge in the queried set, or the number of distinct blocks of the partition present in the queried set, respectively. Our setting is a more constrained case corresponding to queries of size two. Another similar setting is that of *entity resolution* [13] where recent developments also assume perfect oracles and transitive properties using pairwise queries to reconstruct a set partition [23, 25]. In this case, clusters correspond to duplicates of the same entity. Most solutions have a real-valued similarity measure between elements but rely on human verification to improve the results. The entity resolution literature also considers the noisy case for a fixed number of clusters [8, 9, 18, 10].

## 2 Our results

### 2.1 Uniform distribution on partitions

We start by considering the setting where the partition of the $n$-set is chosen uniformly at random among all possible partitions. The *average complexity* of an AC algorithm is the average number of

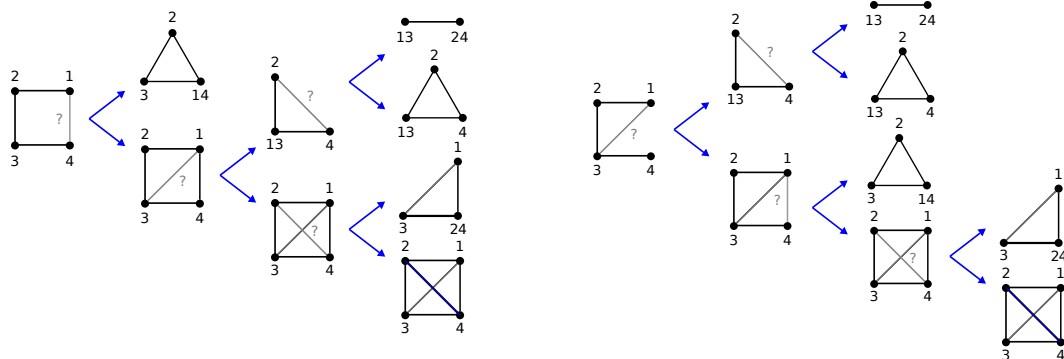

Figure 1: Aggregated graphs obtained after various queries and organized into query trees. The positive answers are displayed on top, the negative ones on the bottom. Left, a non-chordal query, leading to an average complexity of $13/5$. Right, chordal queries from the same initial situation, leading to an optimal average complexity of $12/5$.

queries used to recover the random partition. We will define a class of AC algorithms, called *chordal algorithms*, and prove in Theorem 2 that an algorithm has minimal average complexity if and only if it is chordal. Theorem 3 shows that all chordal algorithms have the same distribution on their number of queries. Finally, this distribution is characterized both exactly and asymptotically in Theorem 4.

It will be useful to use certain graphs marking the 'progress' of the AC algorithm on our $n$-set. Given a partition $P$ of an $n$-set and an AC algorithm, we can associate a graph $G_t$ to each step $t$ of the algorithm. Namely, $G_0$ has $n$ vertices, labeled with the elements of our set, and at time $t \geq 1$, the graph $G_t$ is obtained by adding an edge for each negatively-answered query, and successively merging pairs of vertices that received a positive answer. (A vertex $u$ obtained by joining vertices $v, w$ is adjacent to all neighbors of each of $v, w$, and we label $u$ with the union of its earlier labels.) We call $G_t$ the *aggregated graph* at time $t$. Note that after the last step of the algorithm, the aggregated graph is complete and each of its vertices corresponds to one of the blocks of $P$. Also note that any fixed $G_t$ (with labels on vertices also fixed) may appear as the aggregated graph for more than one partition (possibly at a different time $t'$). We call the set of all these partitions the *realizations* of $G_t$. Those notions are illustrated in Figure 1.

We now need a quick graph-theoretic definition. A cycle $C$ in a graph is *induced* if all edges induced by $V(C)$ belong to the cycle. A graph is *chordal* if its induced cycles all have length three. Chordal graphs appear in many applications. We say that an AC algorithm is *chordal* if one of the following equivalent conditions is satisfied (see the Appendix for a proof of the equivalence of the conditions):

(i) for all input partitions, each aggregated graph is chordal,

(ii) for all input partitions, no aggregated graph has an induced $C_4$,

(iii) for all input partitions, and for every query $u$, $v$ made on some aggregated graph $G_t$, the intersection of the neighborhoods of $u$ and $v$ separates $u$ and $v$ in $G_t$

where a set $S$ (possibly empty) is said to *separate* vertices $u, v \in V(G) \setminus S$ if $u$ and $v$ lie in distinct components of $G - S$. The queries that keep a graph chordal are exactly those satisfying Condition (iii). Thus, it is used in chordal algorithms to identify the candidate queries. On the other hand, a query satisfying Condition (ii) might turn a chordal graph non-chordal by creating an induced cycle of length more than $4$ (in which case, Condition (ii) will fail on a later query). Two examples of chordal algorithms are presented below.

**Definition 1.** *The* clique algorithm *is an AC algorithm that gradually grows the partition, starting with the empty partition. Each new item is compared successively to an item of each block of the partition, until it is either added to a block (positive answer to a query) or all blocks have been visited, in which case a new block is created for this item. The* universal algorithm *finds the block containing the item of largest label by comparing it to all remaining items. The algorithm is then applied to partition the other items, and the previous block is added to the result.*

**Theorem 2.** *On partitions of size $n$ chosen uniformly at random, an AC algorithm has minimal average complexity if and only if it is chordal.*

The *complexity distribution* of an AC algorithm of a set $S$ with $n \geq 1$ elements is a tuple $(a_0, a_1, a_2, \ldots)$ where $a_i$ is the number of partitions of $S$ for which the algorithm has complexity $i$. Clearly, $a_i = 0$ for all $i < n - 1$, and $a_{\binom{n}{2}} = 1$, for any AC algorithm. Our next result shows that constraining the average complexity to be minimal fixes the complexity distribution.

**Theorem 3.** *On partitions of size $n$ chosen uniformly at random, all chordal AC algorithms have the same complexity distribution.*

Note that either of Theorems 2 and 3 implies the weaker statement that all chordal AC algorithms of an $n$-set have the same average complexity, but with very different proofs. Sketches of the proofs of Theorems 2 and 3 can be found in sections 3.1 and 3.2.

In practice, several human experts work in parallel to annotate the training data. The chordal algorithms can easily be parallelized: when an expert becomes available, give him a query that would keep the aggregated graph chordal if all pending queries received negative answers. The condition ensures the chordality of the parallelized algorithm.

Our third theorem for this setting describes the distribution of the number of queries used by chordal algorithms on partitions chosen uniformly at random. Two formulas for the probability generating function are proposed. The first one is well suited for computer algebra manipulations, while the second one, more elegant, is better suited for asymptotic analysis. It is a $q$-analog of Dobiński's formula for Bell numbers

$$B_n = \frac{1}{e} \sum_{m \geq 0} \frac{m^n}{n!}$$

(see *e.g.* [14, p.762]), counting the number of partitions of size $n$. In order to state the theorem, we introduce the $q$-analogs of an integer, the factorial, the Pochhammer symbol and the exponential

$$[n]_q = \sum_{k=0}^{n-1} q^k, \quad [n]_q! = \prod_{k=1}^{n} [k]_q, \quad (a;q)_n = \prod_{k=0}^{n-1} (1 - aq^k), \quad e_q(z) = \sum_{n \geq 0} \frac{z^n}{[n]_q!}.$$

Observe that the $q$-analog reduce to their classic counterparts for $q = 1$. The *Lambert function $W(x)$* is defined for any positive $x$ as the unique solution of the equation $we^w = x$. As $x$ tends to infinity, we have $W(x) = \log(x) - \log(\log(x)) + o(1)$.

**Theorem 4.** *Let $X_n$ denote the complexity of a chordal algorithm on a partition of size $n$ chosen uniformly at random. The distribution of $X_n$ has probability generating function in the variable $q$ equal to the two following expressions*

$$\frac{1}{B_n} \left( \frac{q}{1-q} \right)^n \sum_{k=0}^{n} \binom{n}{k} (-1)^k \left( \frac{1-q}{q}; q \right)_k \qquad and \qquad \frac{1}{B_n} \frac{1}{e_q(1/q)} \sum_{m \geq 0} \frac{[m]_q^n}{[m]_q!} q^{n-m}.$$

*The normalized variable $(X_n - E_n)/\sigma_n$ converges in distribution to a standard Gaussian law, where*

$$E_n = \frac{1}{4}(2W(n) - 1)e^{2W(n)} \qquad and \qquad \sigma_n = \frac{1}{3}\sqrt{\frac{3W(n)^2 - 4W(n) + 2}{W(n) + 1} e^{3W(n)}}.$$

As a corollary, the average complexity of chordal algorithms is asymptotically $\binom{n}{2} \big/ \log(n)$. Because of this almost quadratic optimal complexity, using pairwise queries can only be better than direct classification for relatively small data sets. We arrive at a different conclusion for the model presented in the next section, where AC algorithm have linear complexity.

## 2.2 Random partitions with fixed number of blocks

In many real-world applications, we have information on the number of classes and their respective sizes in the data requiring annotation. This motivates the introduction of the following alternative random model for the set partition investigated by the AC algorithms. Now, the set of partitions of the $n$-set $S$ is distributed in a not necessarily uniform way, but each element of $S$ has a probability of

$p_i$ to belong to the partition class $C_i$ (and the probabilities $p_i$ sum up to 1). Our main focus here will be on the most applicable and accessible variant of the above-described model, where the number of partition class is a fixed number $k$, and $k$ is small compared to $n$. Without loss of generality, we can assume that the probabilities $p_1, p_2, \ldots, p_k$ in this model satisfy $p_1 \geq p_2 \geq \cdots \geq p_k$.

Intuitively, an algorithm with low average complexity should avoid making many comparisons between different classes (one such comparison is always necessary between any pair of classes, but more comparisons are not helpful). For this reason, it seems plausible that an AC algorithm with optimal complexity should compare a new element first with the largest class identified up to this moment, as both the new element and the largest class are most likely to coincide with $C_1$, the 'most likely class'. This is precisely how the clique AC algorithm from Definition 1 operates, where we add the additional refinement that each new item is compared to the blocks discovered so far in decreasing order of their sizes. With this refinement, we conjecture the following.

**Conjecture 5.** *For an $n$-set with a random partition with probabilities $p_1, p_2, \ldots, p_k$, the clique algorithm has minimal average complexity among all AC algorithms.*

In support of Conjecture 5, we present the following results. Firstly, we exhibit the limit behaviour of the expected number of queries of the clique algorithm. The simple proof of this theorem can be found in the appendix. It allows one to decide for practical cases whether direct classification by human experts or pairwise queries will be more efficient to annotate the training data.

**Theorem 6.** *Let $p_1 \geq \cdots \geq p_k$ be fixed with $\sum_{i=1}^{k} p_i = 1$. Let $X$ denote the expected number of queries made by the clique algorithm with these parameters for an $n$-set. Then $\mathbb{E}[X] \sim \sum_{i=1}^{k} i p_i n$.*

We now compare the complexity of the clique algorithm with the complexity of an algorithm that chooses its queries randomly. More precisely, we define the *random algorithm* to be the one that at each step $t$ compares a pair of elements chosen uniformly at random among all pairs whose relation is not known at this time (that is, they are neither known to belong to the same class, nor known to belong to different classes). The reason for analyzing the random algorithm is that one may think of this algorithm as the one that models the situation where no strategy at all is used by the human experts, which might make this procedure cheap to implement. It turns out, however, that there is a cost in form of larger average complexity associated to the random algorithm, if compared to our proposed candidate for the optimal algorithm, the clique algorithm.

**Theorem 7.** *Let $p_1 \geq \cdots \geq p_k$ be fixed with $\sum_{i=1}^{k} p_i = 1$. Let $X$ be the number of queries of a random algorithm with these parameters for an $n$-set. Then $\mathbb{E}[X] \sim n - k + \sum_{i<j} f(p_i, p_j) n$, where*

$$f(\alpha, \beta) = \begin{cases} 2\alpha & \alpha = \beta \\ \frac{2\alpha\beta \ln(\alpha/\beta)}{\alpha - \beta} & \alpha \neq \beta. \end{cases}$$

Note that if all the $p_i$ are equal to $1/k$, then by Theorems 6 and 7, the expected number of queries produced by the random algorithm is asymptotically $n - k + (2n/k)\binom{k}{2} = k(n-1)$, while for the clique algorithm this number is $(k+1)n/2$.

## 2.3  Noisy queries

We now discuss the situation when answers to the query can be inconsistent, due to errors from the human experts or ambiguity in the data. Different perspectives can be found in [10, 18]. Let $G$ denote the graph where vertices correspond to items from the data set, edges to queries, so each edge is either *positive* or *negative* depending on the answer. Because of potential errors, the aggregated graph is not sufficient for the analysis anymore, but it can be recovered by contracting to a vertex each positive component of $G$.

**Correcting errors.**   An inconsistency is detected if and only if $G$ contains a *contradictory cycle*: a cycle where all edges except one are positive. At each step, we consider the shortest contradictory cycle and ask a query that cuts it in two, following a divide-and-conquer strategy. After a logarithmic number of queries, if no additional error occurred, a false answer or ambiguous data is identified and the answer is corrected. At any point, if the number of contradictions detected grows out of proportion (edit war between human experts), a classic clustering algorithm can be applied as a last resort to settle the differences of opinions. We now focus on the problem of detecting inconsistencies.

**Bounded number of errors.** To ensure the detection of at most $k$ errors, each positive component of graph $G$ must be $(k+1)$-edge-connected and any two positive components must be linked by at least $k+1$ negative edges. The minimal number of queries for $n$ items and $b$ blocks, assuming no block has size 1, is then $(k+1)\big(\binom{b}{2} + n/2\big)$, because each vertex of $G$ has at least $k+1$ positive neighbors.

**Small probability of error.** To avoid this high cost, the error detection criteria can be relaxed. We now consider that each answer has a small probability $p$ of error and add a few queries at the end of the AC algorithm, while keeping the probability of undetected errors low. More precisely, let $c_0 + c_1 p + c_2 p^2 + \cdots$ denote the Taylor expansion of this probability at $p = 0$. Since $p$ is assumed to be small, our aim is to minimize the vector $(c_0, c_1, \ldots)$ for the lexicographic order. During the AC algorithm, when a query between two classes is needed, we choose the items in those classes to maintain the following structure. Any positive component should be a tree with vertices of degree at most 3. We call 2-*path* a path of vertices of degree 2 linking two vertices of degree 3. The 2-paths should all have length close to a parameter $r$ of the algorithm. At the end of the algorithm, we introduce additional queries between pairs of leaves of the same tree, turning each positive component of size at least 2 into either a 3-edge-connected graph or a cycle of length close to $r$. Queries are also added to ensure that any two positive components are linked by at least 3 negative edges. Assuming the partition contains $b$ blocks and the average length of the 2-paths is $r$, the number of additional queries compared to the noiseless setting is approximately $\frac{n}{3r+2} + b$ inside the blocks, plus the potential queries between blocks (at most $2\binom{b}{2}$). This setting ensures $c_0 = c_1 = 0$ and minimizes $c_2$ (see [6]). If the length of all 2-paths and the sizes of positive components that are cycles are bounded by $r'$, then $c_2 = \binom{r'+1}{2}\frac{3n}{3r+2}$. Thus, the choice of the parameter $r$ is a trade-off between the number of additional queries and the robustness to noise of the algorithm.

## 3 Proofs

### 3.1 Proof of Theorem 2

We will prove Theorem 2 by analyzing the types of queries made by an AC algorithm. For this, we classify the queries made by an AC algorithm into three types:

- *Core* queries are those that receive a positive answer.
- A query at time $t$ is *excessive* if it compares vertices $x$ and $y$ that are joined by an induced path in $G_t$ on an even number of vertices that alternates between two partition classes.
- A query at time $t$ is *productive* if it is neither core nor excessive.

Note that each query is of exactly one type, as excessive queries have to receive negative answers. Also note that as the core queries are the only ones that shrink $G_t$, the number of core queries does not depend on the algorithm.

**Lemma 8.** *For any AC algorithm and any partition of a set of size $n$ containing $k$ classes, the number of core queries is exactly $n - k$.* □

Next, we show that for a random partition, also the number of productive queries is, in expectation, the same for all AC algorithms.

**Lemma 9.** *Let $\mathcal{P}_k$ be the set of partitions of $[n]$ into exactly $k$ sets, and choose $P \in \mathcal{P}_k$ uniformly at random. Then all algorithms have the same expected number of productive queries.*

Because of space constraints, we only present a sketch of the proof of this lemma here. The full proof of Lemma 9 is in the appendix.

*Sketch of the proof of Lemma 9.* Consider a partition of $[n]$ into $k$ sets $C_1, C_2, \ldots, C_k$, and let $q_{ij}$ be the number of productive queries comparing a vertex from $C_i$ with a vertex from $C_j$. Then $\mathbb{E}[q_{ij}] = \mathbb{E}[q_{12}]$ for all $i, j$, and by linearity of expectation the expected number of productive queries is $\binom{k}{2}\mathbb{E}[q_{12}]$. Thus, we are done if we can prove that $\mathbb{E}[q_{12}]$ is independent from the choice of the algorithm. It suffices to show that $\mathbb{E}[q_{12}|C_1 \cup C_2 = S]$ is independent from the choice of algorithm

because

$$\mathbb{E}[q_{12}] = \sum_{S \subset [n], |S| \geq 2} \mathbb{E}[q_{12}|C_1 \cup C_2 = S]\mathbb{P}[C_1 \cup C_2 = S].$$

In other words, we wish to understand the expected number of productive queries of an AC algorithm working on a partition with exactly two classes of a set of size $|S|$, chosen uniformly at random (there are $2^{|S|} - 2$ such partitions). Observe that this is very similar to calculating the expected number of productive queries for a uniformly-chosen partition with *at most* two classes (there are $2^{|S|}$ such partitions). In fact, it it not hard to see that these numbers only differ by a factor of $\frac{2^{|S|}}{2^{|S|-2}}$. So we can restrict our attention to the expectation of the latter number, which is easier to calculate, as the model is equivalent to assigning independently to each element of $S$ a value in $\{0, 1\}$ uniformly at random.

More precisely, we will now argue that the expected number of productive queries in this scenario is $\frac{n-1}{2}$, for each partition algorithm. For this, note that each component of any aggregated graph $G_t$ is either a single vertex or a nonempty bipartite graph. Moreover, each component has two possible colorings, and it is not hard to show by induction that these colorings are equally likely. So, whenever the algorithm makes a query for two vertices from distinct components, answers 'yes' and 'no' are equally likely. As the algorithm makes $n - 1$ such queries (since each such query reduces the number of components by 1), it follows from linearity of expectation, that the expected number of productive queries is $(n - 1)/2$. $\square$

Finally, we analyze the excessive queries of an AC algorithm.

**Lemma 10.** *An AC algorithm makes no excessive queries if and only if it is chordal.*

For the proof, we need the following easy lemma whose proof can be found in the appendix.

**Lemma 11.** *For each non-chordal AC algorithm, there is an input partition and a time $t$ such that $G_t$ has an induced $C_4$ one of whose edges comes from a negative query in step $t - 1$.*

*Proof of Lemma 10.* By definition, a chordal AC algorithm has no aggregated graphs with induced cycles of length at least $4$. So, since an excessive query, if answered negatively, creates an induced cycle of even length at least $4$, chordal algorithms make no excessive queries.

For the other direction, let us show that any non-chordal AC algorithm makes at least one excessive query for some partition. By Fact 11, a non-chordal AC algorithm has, for some partition of the ground set, an aggregated graph $G_t$ with an induced $C_4$, on vertices $v_1, v_2, v_3, v_4$, in this order, such that the last query concerned the pair $v_1, v_4$. There is a realization of $G_t$ where $v_1$ and $v_3$ are in one partition class, and $v_2$ and $v_4$ are in another partition class. For this partition, the query $v_1, v_4$ at time $t - 1$ is an excessive query. $\square$

We are ready for the proof of Theorem 2.

*Proof of Theorem 2.* By Lemmas 8 and 9, all AC algorithms have the same expected number of core queries and productive queries. So the optimal algorithms are the ones with the minimum expected number of excessive queries. By Lemma 10, these are the chordal algorithms. $\square$

### 3.2 Proof of Theorem 3

The full proof is in the appendix. We prove a more general result which allows for the algorithm to start with any aggregated graph instead of the empty graph. We prove this using induction on the number of non-edges of $G = (V, E)$, the base case being trivial. We fix some useful notation now: For $x, y \in V$, set $G(xy) = (V, E \cup \{xy\})$, and let $G_{xy}$ be obtained from $G$ by identifying $x$ and $y$.

For the induction step consider a graph $G$ with $k + 1$ missing edges, and let $A_0$, $A_1$ be two distinct chordal AC algorithms for $G$. By induction, we can assume that $A_0$ and $A_1$ differ in their first queries. Say the first query of $A_i$ is $u_i$, $v_i$, for $i = 0, 1$. Then $G(u_iv_i)$ is chordal for $i = 0, 1$. Note that we can assume that $u_0 \neq u_1$. We distinguish two cases.

**Case 1.** $G(u_0v_0)(u_1v_1)$ is chordal and moreover, if $v_0 = v_1$ then $u_0u_1 \notin E(G)$. Then, for $i = 0, 1$, the edge $u_iv_i$ can be chosen as the first edge of a chordal AC algorithm for $G(u_{1-i}v_{1-i})$ or for

$G_{u_{1-i}v_{1-i}}$. As the induction hypothesis applies to $G(u_{1-i}v_{1-i})$ and to $G_{u_{1-i}v_{1-i}}$, we can assume that $u_iv_i$ is the second edge in $A_{1-i}$. Observe that for each $i = 0, 1$ after the second query of $A_i$, we arrive at one of the four graphs $(G_{u_0v_0})_{u_1v_1}, G(u_0v_0)_{u_1v_1}, G(u_1v_1)_{u_0v_0}, G(u_0v_0)(u_1v_1)$. Thus the complexity distribution of $A_0$ and $A_1$ is identical (as it can be computed from the complexity distribution for the algorithms starting at these four graphs).

**Case 2.** $G(u_0v_0)(u_1v_1)$ is chordal, $v_0 = v_1$ and $u_0u_1 \in E(G)$, or $G(u_0v_0)(u_1v_1)$ is not chordal. This case is harder to analyze, but one can show that either there is an edge $uv \in E(G)$ such that $G(uv)$, $G(uv)(u_0v_0)$ and $G(uv)(u_1v_1)$ are chordal, or $G$ has a very specific shape. In the former case, we proceed as in the previous paragraph using the edge $uv$ as a proxy. In the latter case our argument relies on our knowledge on the structure of $G$.

### 3.3 Proof of Theorem 4

According to Theorem 3, all chordal algorithms share the same complexity distribution, so we investigate a specific chordal algorithm, the *universal AC algorithm* (see Definition 1) without loss of generality. This algorithm first computes the block $B$ containing the largest item by comparing it to all other items, then calculates the partition $Q$ for the remaining items, and finally inserts the block $B$ in $Q$. Let $\mathrm{query}(p)$ denote the number of queries used by the universal AC algorithm to recover partition $p$. Let us introduce the generating function

$$P(z, q) = \sum_{\text{partition } p} q^{\mathrm{query}(p)} \frac{z^{|p|}}{|p|!}.$$

The *symbolic method* presented in [14] translates the description of the algorithm into a the differential equation characterizing $P(z, q)$

$$\partial_z P(z, q) = P(qz, q)e^{qz}$$

with initial condition $P(0, q) = 1$. Observing that the function $e^{\frac{q}{1-q}z}$ is solution of a similar differential equation, we consider solutions of the form $P(z, q) = A(z, q)e^{\frac{q}{1-q}z}$. The differential equation on $P(z, q)$ translates into a differential equation on $A(z, q)$

$$\partial_z A(z, q) + \frac{q}{1 - q}A(z, q) = A(qz, q)$$

with initial condition $A(0, q) = 1$. Decomposing $A(z, q)$ as a series in $z$, we find the solution

$$A(z, q) = \sum_{k \geq 0} \left(\frac{1-q}{q}; q\right)_k \frac{\left(-\frac{q}{1-q}z\right)^k}{k!}.$$

By definition, the probability generating function $\mathrm{PGF}_n(q)$ of the complexity distribution when the set of items has size $n$ is linked to our generating function by the relation

$$\mathrm{PGF}_n(q) = \frac{n!}{B_n}[z^n]P(z, q),$$

where the Bell number $B_n$ counts the number of partitions of size $n$. The first exact expression from the theorem is obtained directly by coefficient extraction. The second one requires using the following classic $q$-identities

$$[n]_q! = \frac{(q; q)_n}{(1 - q)^n}, \quad \frac{1}{(x; q)_\infty} = \sum_{n \geq 0} \frac{x^n}{(q; q)_n}, \quad e_q(x) = ((1 - q)x; q)_\infty^{-1}.$$

To obtain the Gaussian limit law, we prove that the Laplace transform of the normalized random variable $X_n^\star = (X_n - E_n)/\sigma_n$

$$\mathbb{E}(e^{sX_n^\star}) = \mathrm{PGF}_n(e^{s/\sigma_n})e^{-sE_n/\sigma_n}$$

converges to the Laplace transform of the standard Gaussian $e^{s^2/2}$ pointwise for $s$ in a neighborhood of 0. To do so, we apply to the second expression the Laplace method for sums [14, p. 761], using in the process a $q$-analog of Stirling's approximation [20].

### 3.4 Proof of Theorem 7

Since Lemma 8 is still valid in this setting, it suffices to prove the following, with $f$ as in Theorem 7.

**Theorem 12.** *Let $p_1 \geq \cdots \geq p_k$ be fixed with $\sum_{i=1}^{k} p_i = 1$. Let $X$ denote the number of edges between classes using a random algorithm. Then $\mathbb{E}[X] \sim \sum_{i<j} f(p_i, p_j)n$.*

*Proof.* We only give a sketch here, see the appendix for further detail. Instead of analyzing the random algorithm, we analyze the following process. Begin with all vertices marked *active*. At each time step, pick (with replacement) a random pair $\{u, v\}$ and:

- If $u, v$ are active and from distinct classes $i, j$ then say we have generated an $ij$-*crossedge*.

- If $u, v$, are both active and in class $i$ then mark exactly one of $u$ and $v$ inactive.

- If one of $u, v$ is inactive, then do nothing.

Note that as the process runs, the number of active vertices is monotonic decreasing, and we are increasingly likely to choose pairs where one vertex is inactive. These contribute to the new process, but do not generate new comparisons between pairs. So we are looking at a (randomly) slowed down version of the random algorithm; but this makes the analysis much simpler.

Let $x_i(t)$ denote the number of active class $i$ vertices after $t$ time steps, and $x_{ij}(t)$ denote the number of $ij$-crossedges that are generated in the first $t$ time steps. Then at step $t + 1$, the probability that we pick two active vertices in class $i$ is

$$\binom{x_i(t)}{2} \Big/ \binom{n}{2} \sim \frac{x_i(t)^2}{n^2}.$$

Writing $p = p_i$, we estimate $x_i(t)$ via a function $x = x(t)$ satisfying the differential equation $\partial_t x(t) = -\frac{x(t)^2}{n^2}$, with $x(0) = pn$ This has solution

$$x(t) = \frac{n^2}{t + n/p} = \frac{pn}{1 + pt/n}.$$

One can show that with high probability, $x_i(t)$ tracks $x(t)$ quite closely.

Now we estimate the number of $ij$-crossedges. Using our estimates for $x_i(t)$ and $x_j(t)$, we see that the probability of an $ij$ crossedge at step $t + 1$ is

$$x_i(t)x_j(t) \Big/ \binom{n}{2} \sim \frac{2x_i(t)x_j(t)}{n^2} \approx \frac{2n^2}{(t + n/p_i)(t + n/p_j)}.$$

Let $p = p_i$ and $q = p_j$. Similarly as before, we can model the growth of $x_{ij}(t)$ by a function $c = c(t)$ satisfying the differential equation

$$\partial_t c(t) = \frac{2n^2}{(t + n/p)(t + n/q)}$$

with $c(0) = 0$. Calculations show that if $p = q$, then

$$c(t) = 2pn \left(1 - \frac{1}{2 + 2pt/n}\right) \sim 2pn$$

and if $p \neq q$, then

$$c(t) \sim \frac{2npq \ln(p/q)}{p - q}$$

as $t/n \to \infty$. In order to prove the theorem, we now run the process for time $Kn$, where $K$ is a large constant. We note that at this point, we have (with high probability) at most $cn$ remaining active vertices for some small constant $c$. But now we revert to the original process: noting that for any partition into $k$ classes, a fraction of at least (about) $1/k$ of the pairs lie inside some class, we see that at each step the process reduces the number of vertices with constant probability. The remaining expected running time is therefore $O(kcn)$. $\qquad \square$

We note that it is also possible to prove a central limit theorem for the running time of the clique algorithm (roughly: after $n^{1/2}$ comparisons, we are very likely to have seen representatives from every set in the partition, and more from the $i$th class than the $j$th class whenever $p_i > p_j$; we estimate the contribution from the remaining steps by an sum of independent random variables). We do not pursue the details here.

## 4 Conclusion

In this article, motivated by the building of training data for supervised learning, we studied active clustering using pairwise comparisons and without any other additional information (such as a similarity measure between items).

Two random models for the secret set partition where considered: the uniform model and the bounded number of blocks model. They correspond to different practical annotation situations. The uniform model reflects the case where nothing is known about the set partition. In that case, many clusters will typically be discarded at the end of the clustering process, as they are too small for the supervised learning algorithm to learn anything from them. Thus, this is a worst case scenario. When some information is available on the set partition, such as its number of blocks or their relative sizes, the bounded number of blocks model becomes applicable.

**Comparison between direct labeling and pairwise comparisons.** As a practical application of this work, we provided tools to decide whether direct labeling or pairwise comparisons will be more efficient for a given annotation task. One should first decide whether the uniform model or the bounded number of blocks model best represents the data. In both cases, our theorems provide estimates for the number of pairwise queries required. Then the time taken by an expert to answer a direct labeling query or a pairwise comparison should be measured. Finally, combining those estimates, the time required to annotate the data using direct labeling or pairwise comparison can be compared. We also provided tools to detect and correct errors in the experts' answers.

**Similarity measures.** Generally, a similarity measure on the data is available and improves the quality of the queries we can propose to the human experts. This similarity measure can be a heuristic that depends only on the format of the data. For example, if we are classifying technical documents, a text distance can be used. The similarity could also be trained on a small set of data already labeled. This setting has been analyzed by [19]. Our motivation for annotating data is to train a supervised learning algorithm. However, one could use active learning to merge and replace the annotation and learning steps. The problem is then to find the queries that will improve the learned classifier the most.

In this article, we focused on the case where such similarity measures are not available. However, we are confident that the mathematical tools developed here will be useful to analyze more elaborate settings as well. In particular, the aggregated graph contains exactly the information on the transitive closure of the answers to the pairwise queries, so its structure should prove relevant whenever pairwise comparisons are considered.

**Open problems.** We leave as open problems the proof that the clique algorithm reaches the minimal average complexity in the bounded number of blocks model, and the complexity analysis of the random algorithm in the uniform model.

## Acknowledgments and Disclosure of Funding

We thank Maria Laura Maag (Nokia) for introducing us to the problem of clustering using pairwise comparisons, and Louis Cohen, who sadly could not continue working with us on this topic. The authors of this paper met through and were supported by the RandNET project (Rise project H2020-EU.1.3.3). Alex Scott was supported by EPSRC grant EP/V007327/1. Quentin Lutz and Élie de Panafieu produced part of this work at Lincs (www.lincs.fr). Quentin Lutz is supported by Nokia Bell Labs (CIFRE convention 2018/1648). Maya Stein was supported by ANID Regular Grant 1180830, and by ANID PIA ACE210010.

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
