# A   Appendix

## A.1   Equivalence in the definition of chordal algorithms

In this short section we show the equivalence of the conditions in the definition of chordal AC algorithms from section 2.1, and some related easy lemmas. We start by proving a lemma from section 3.1.

*Proof of Lemma 11.*  Since the algorithm is not chordal, for some input partition one of the aggregated graphs $G_t$ of the AC algorithm has an induced cycle $C$ of length at least $4$. Consider a realization of $G_t$ where the vertices of $C$ are all in distinct partition classes. Then there is a time $t'$ when four of the vertices of $C$ form an induced $C_4$ in $G_{t'}$.  □

The next lemma is used both in the proof of the equivalence of the conditions in the definition of chordal AC algorithms, and in section A.3. Recall that for a graph $G = (V, E)$ we write $G(uv) = (V, E \cup \{uv\})$.

**Lemma 13.** *Let $G$ be a chordal graph, let $u, v \in V(G)$ be distinct and nonadjacent, and assume $G(u, v)$ is chordal. Then $N(u) \cap N(v)$ is complete and separates $u$ and $v$ (in $G$).*

*Proof.*  Note that $N(u) \cap N(v)$ is complete, as otherwise there are two non-adjacent vertices $x, y \in N(u) \cap N(v)$, and $(u, x, v, y)$ is an induced cycle in $G$, a contradiction. It remains to show that $N(u) \cap N(v)$ separates $u$ from $v$. Assume this is not the case, then there is an induced path $P = (u, p_1, ..., p_k, v)$ that avoids $N(u) \cap N(v)$. In particular, $k \geq 2$. Adding the edge $uv$ to $P$ we obtain an induced cycle of length at least $4$, a contradiction to $G(u, v)$ being chordal.  □

Now we prove the equivalence of the three conditions.

**Lemma 14.** *The following conditions are equivalent for any AC algorithm.*

 (i) *for all input partitions, each aggregated graph is chordal,*

 (ii) *for all input partitions, no aggregated graph has an induced $C_4$,*

 (iii) *for all input partitions, for each query $u$, $v$, the intersection of the neighborhoods of $u$ and $v$ separates $u$ and $v$.*

*Proof.*  By Lemma 11, we know that (ii) implies (i), and by Lemma 13 we know that (i) implies (iii). So we only need to show that (iii) implies (ii). For this, assume there is an input partition, and a time $t$, such that the aggregated graph $G_t$ at time $t$ has an induced cycle $(v_1, v_2, v_3, v_4)$. We can assume that $t$ is the first such time. Then either $G_t$ arose by adding an edge of this cycle, say the edge $v_1 v_4$, or by identifying two vertices $x$ and $y$ to a new vertex from the cycle, say $v_1$. In the first case, the query $v_1$, $v_4$ in step $t - 1$ did not meet the requirement in (iii), a contradiction. In the second case, $x$ and $y$ are joined by the induced path $(x, v_2, v_3, v_4, y)$ or $(y, v_2, v_3, v_4, x)$, so the query $x$, $y$ in step $t - 1$ did not meet the requirement in (iii), a contradiction.  □

## A.2   Full proof of Lemma 9

Let $\mathcal{P}_2(n)$ denote the family of partitions of size $n$ where each element receives a random value in $\{0, 1\}$ uniformly and independently, so that two elements belong to the same block if and only if they share the same value. Thus, all partitions in $\mathcal{P}_2(n)$ have one or two blocks.

**Lemma 15.** *The expected number of productive queries made by any AC algorithm on a random partition from $\mathcal{P}_2(n)$ is exactly $\frac{n-1}{2}$.*

*Proof.*  Let us consider the (random) sequence of graphs $G_1, \ldots, G_n$ produced by running the algorithm on a random element of $\mathcal{P}_2(n)$. As core queries result in contractions, it follows that (for every $t$) every component of $G_t$ is either a single vertex or a nonempty bipartite graph. Note that for each component there are two possible colorings; thus the number of possible colorings of $G_i$ is $2^{\kappa(G_i)}$, where $\kappa$ denotes the number of components.

We prove by induction that:

- For each $i$, if the $i$th query joins two components of $G_{i-1}$ then it is productive with probability half.

- The $2^{\kappa(G_i)}$ colorings of $G_i$ are equally likely.

The second property holds for $G_0$, so it is enough to prove the inductive step. Consider the query made at stage $t$, say joining vertices $x$ and $y$. If $x$ and $y$ lie in the same component of $G_{t-1}$ then the query is excessive, the components do not change, and the two bullets hold. Thus we may restrict our attention to queries such that $x$ and $y$ lie in distinct components, say $H_x$ and $H_y$. There are four possible colorings of $H_x \cup H_y$. The four colorings are equally likely and are independent from the coloring of the rest of the graph. It is now easily checked that $x$ and $y$ have the same color with probability $1/2$, and so the probability that the query is productive is $1/2$. Furthermore, adding the edge $xy$ joins $H_x$ and $H_y$ into a single component, and the two possible colorings of this component are equally likely (and remain independent from the coloring of the rest of $G$). Thus the two bullets hold, and the induction is complete.

There are in total $n-1$ steps at which the algorithm makes a query joining distinct components (as each such query reduces the number of components by 1). So, by linearity of expectation, the expected number of productive queries is $(n-1)/2$. $\qquad\square$

**Lemma 16.** *The expected number of productive queries of an AC algorithm working on a partition of a set $S$ containing exactly two blocks chosen uniformly at random is exactly*

$$\frac{2^{|S|}}{2^{|S|}-2} \frac{|S|-1}{2}.$$

*Proof.* Consider an AC algorithm, and let $\alpha(|S|)$ denote the expected number of productive queries. Now run the algorithm on a partition from $\mathcal{P}_2(|S|)$. If the algorithm is fed one of the two constant colorings then it makes exactly $|S|-1$ queries, all of which are core (and therefore not productive).

The probability that a partition $P \in \mathcal{P}_2(n)$ is constant is $2/2^{|S|}$; and if we condition on $P$ being nonconstant then it is uniformly distributed among the set of partitions with exactly two blocks. By Lemma 15, we conclude that the expected number of productive queries satisfies

$$\frac{|S|-1}{2} = \alpha(|S|)\mathbb{P}[P \text{ nonconstant}] + 0\mathbb{P}[P \text{ constant}] = \frac{2^{|S|}-2}{2^{|S|}}\alpha(|S|).$$

The result follows by solving for $\alpha(|S|)$. $\qquad\square$

We are now ready for the full proof of Lemma 9.

*Proof of Lemma 9.* Fix $k$ and consider a partition of $[n]$ into exactly $k$ sets $C_1, C_2, \ldots, C_k$. Let $\alpha_{ij}$ denote the expected number of productive queries that compare a vertex from $C_i$ with a vertex from $C_j$. Then all $\alpha_{ij}$ are equal, and by linearity of expectation the expected number of productive queries is $\binom{k}{2}\alpha_{12}$. Thus it is enough to prove that $\alpha_{12}$ does not depend on the choice of algorithm.

Let $q_{12}$ be the number of productive queries comparing a vertex from $C_1$ with a vertex from $C_2$. (so $\alpha_{12} = \mathbb{E}q_{12}$) Then, considering the set $S = C_1 \cup C_2$, and applying Lemma 16, we obtain

$$\alpha_{12} = \sum_{S \subset [n], |S| \geq 2} \mathbb{E}[q_{12}|C_1 \cup C_2 = S]\mathbb{P}[C_1 \cup C_2 = S]$$

$$= \sum_{S \subset [n], |S| \geq 2} \frac{|S|-1}{2} \frac{2^{|S|}}{2^{|S|}-2}\mathbb{P}[C_1 \cup C_2 = S]. \qquad (1)$$

The last line is independent from the choice of algorithm, which concludes the proof. $\qquad\square$

## A.3 Proof of Theorem 3

This section is devoted to the proof of Theorem 3. For this we will need several auxiliary lemmas. We also recall a useful notation introduced earlier: For a graph $G = (V, E)$ let $G(uv) = (V, E \cup \{uv\})$, and let $G_{uv}$ be the graph obtained from $G$ by identifying $u$ and $v$.

The first of our lemmas shows that every chordal graph can 'grow' an edge while staying chordal.

**Lemma 17.** *Let $H$ be a chordal graph that is not complete. Then $H$ has a non-edge $e$ such that $H(e)$ is chordal.*

*Proof.* Let $u$ be a non-universal vertex of $H$. Among all non-neighbors of $u$, choose $p_1$ such that $|N(u) \cap N(p_1)|$ is maximized. We claim that $H(up_1)$ is chordal.

Indeed, otherwise there is an induced cycle $C = (u, p_1, \ldots, p_k, u)$, with $k \geq 3$. As $p_k \in N(u) \cap N(p_{k-1}) \setminus N(p_1)$, our choice of $p_1$ guarantees that there is a vertex $w \in N(u) \cap N(p_1) \setminus N(p_{k-1})$. Let $j \leq k - 2$ be the largest index in $[k-2]$ such that $wp_j \in E(H)$. Then, depending on whether the edge $wp_k$ is present, either $(w, p_j, \ldots, p_k, u, w)$ or $(w, p_j, \ldots, p_k, w)$ is an induced cycle of length at least 4 in $H$, a contradiction. □

Our next two lemmas are more technical. They give a structural characterization of those aggregated graphs where consecutive queries cannot easily be interchanged. In order to make their statement easier, let us say that a graph $G$ has a *complete separation* $(A, K, B)$ if $V(G)$ is the disjoint union of $A, B, K$ so that $A \neq \emptyset \neq B$, each of $A \cup K$ and $B \cup K$ is complete, and there are no edges from $A$ to $B$. (Observe that we allow $K$ to be empty.)

**Lemma 18.** *Let $G$ be a chordal graph that does not have a complete separation. For $i = 0, 1$ let $u_i v_i$ be a non-edge of $G$ such that $G(u_i v_i)$ is chordal, and $G(u_0 v_0)(u_1 v_1)$ is non-chordal. Then there is a non-edge $uv$ of $G$ such that $G(uv)$ is chordal, and $G(u_i v_i)(uv)$ is chordal for $i = 0, 1$.*

**Lemma 19.** *Let $G$ be a chordal graph that does not have a complete separation. Let $u_0, u_1, v \in V(G)$ such that for $i = 0, 1$, we have $u_i v \notin E(G)$ and $G(u_i v)$ is chordal, and moreover, $u_0 u_1 \in E(G)$. Then there is a non-edge $uw$ of $G$ such that $G(uw)$ is chordal, and $G(u_i v_i)(uw)$ is chordal for $i = 0, 1$.*

The proofs of these two lemmas rely on purely structural arguments and will be postponed to the end of the section.

We are now ready to prove Theorem 3. Actually, we will prove a more general result which allows for the algorithm to start with any aggregated graph instead of starting with the empty graph. More precisely, if $G$ is an aggregated graph at time $t$ for some AC algorithm for an $n$-set $S$, then we call the restriction of the algorithm to all queries after time $t$ that eventually lead to a realization of $G$ a *AC algorithm starting at $G$*. We define the *complexity distribution* of this algorithm analogously to our earlier definition. In particular, if $G$ is complete, then the complexity distribution is $(a_0, a_1, a_2, \ldots, a_{\binom{n}{2}}) = (1, 0, 0, \ldots, 0)$.

**Theorem 20.** *For any $G$, all chordal AC algorithms starting at $G$ have the same complexity distribution.*

*Proof.* We proceed by induction on the number of non-edges of $G$. Proving the base case is trivial as, starting from a complete graph, the only AC algorithm has distribution $(a_0, a_1, a_2, \ldots, a_{\binom{n}{2}}) = (1, 0, 0, \ldots, 0)$.

For the induction step assume that for any chordal graph with $k$ or less missing edges, all chordal AC algorithms have the same complexity distribution, and consider a graph $G$ with $k + 1$ missing edges. Let $A_0$, $A_1$ be two distinct chordal AC algorithms for $G$. If their first queries are the same, say they query the edge $e$, then by induction we know that for both $G_e$ and $G(e)$, the two algorithms have the same distribution if we let them start there. As the distribution for an algorithm starting at $G$ is uniquely obtained from the complexity distributions of the same algorithm starting at $G_e$ and at $G(e)$, we see that $A_0$ and $A_1$ have the same complexity distribution.

So we can assume that $A_0$ and $A_1$ differ in their first queries. Say the first query of $A_i$ is $u_i, v_i$, for $i = 0, 1$. Then $G(u_i v_i)$ is chordal for $i = 0, 1$. Note that we can assume that $u_0 \neq u_1$. We will distinguish two cases.

First, let us assume that $G(u_0 v_0)(u_1 v_1)$ is chordal and moreover, if $v_0 = v_1$ then $u_0 u_1 \notin E(G)$. Then, for $i = 0, 1$, the edge $u_i v_i$ can be chosen as the first edge of a chordal AC algorithm for $G(u_{1-i} v_{1-i})$ or for $G_{u_{1-i} v_{1-i}}$. As the induction hypothesis applies to $G(u_{1-i} v_{1-i})$ and to $G_{u_{1-i} v_{1-i}}$, we can assume that $u_i v_i$ is the second edge in $A_{1-i}$. Observe that for each $i = 0, 1$ after the second query of $A_i$, we arrive at one of the four graphs $(G_{u_0 v_0})_{u_1 v_1}, G(u_0 v_0)_{u_1 v_1}, G(u_1 v_1)_{u_0 v_0}, G(u_0 v_0)(u_1 v_1)$.

Thus the complexity distribution of $A_0$ and $A_1$ is identical (as is can be computed from the complexity distribution for the algorithms starting at these four graphs).

Now, let us assume that either $G(u_0v_0)(u_1v_1)$ is chordal, $v_0 = v_1$ and $u_0u_1 \in E(G)$, or $G(u_0v_0)(u_1v_1)$ is not chordal. Then, by Lemmas 18 and 19, we know that either there is an edge $uv \in E(G)$ such that $G(uv)$, $G(uv)(u_0v_0)$ and $G(uv)(u_1v_1)$ are chordal, or $V(G)$ can be partitioned into three sets, $A$, $B$ and $K$, such that $A \cup K$ and $B \cup K$ are complete and $K$ separates $A$ from $B$. If the former is the case, we can proceed as in the previous paragraph to see that every chordal AC algorithm starting with $u_0v_0$ has the same complexity distribution as any of the chordal AC algorithms starting with $uv$ (note that such algorithms exist by Lemma 17), which, in turn, has the same complexity distribution as any of the chordal AC algorithms starting with $u_1v_1$, leading to the desired conclusion.

So assume there are sets $A$, $B$ and $K$ as above. By symmetry, we can assume that $u_0, u_1 \in A$ and $v_0, v_1 \in B$. Consider the automorphism $\sigma$ of $G$ that maps $u_0$ to $u_1$ and $v_0$ to $v_1$ while keeping all other vertices fixed. We can now view $A_0$ as an algorithm in $\sigma(G)$ that starts with the edge $u_1v_1$. By the induction hypothesis, we conclude that $A_1$ (for $G$) has the same distribution as $A_0$ (for $\sigma(G)$, and thus also for $G$). $\qquad\square$

It remains to prove Lemmas 18 and 19. We start by giving a characterization of graphs that remain chordal when we add either one of two edges, but not if we add both.

**Lemma 21.** *Let $G$ be a chordal graph and let $u_0, u_1, v_0, v_1 \in V(G)$ such that for $i = 0, 1$, vertices $u_0, u_1, v_i$ are all distinct, $u_iv_i \notin E(G)$, and $G(u_iv_i)$ is chordal. If $G(u_0v_0)(u_1v_1)$ is not chordal, then $K := N(u_0) \cap N(u_1) \cap N(v_0) \cap N(v_1)$ is complete, and $G - K$ has two distinct components $A$ and $B$, such that either $u_0, u_1 \in A$ and $v_0, v_1 \in B$, or $u_0, v_1 \in A$ and $u_1, v_0 \in B$.*

*Proof.* As $G(u_0v_0)(u_1v_1)$ is not chordal, we know that $G(u_0v_0)(u_1v_1)$ has an induced cycle $C = (u_1, v_1, ..., v_{\ell-1}, v_\ell, ..., v_k, u_1)$, with $k \geq \ell \geq 2$, where either $v_{\ell-1} = v_0$ and $v_\ell = u_0$, or $v_{\ell-1} = u_0$ and $v_\ell = v_0$. According to Lemma 13, since $G(u_iv_i)$ is chordal, the set $K_i := N(u_i) \cap N(v_i)$ is complete and separates $u_i$ and $v_i$ in $G$, for each $i \in \{0, 1\}$. Since $C$ has at least four vertices, and $K_i$ has neighbors $u_i, v_i$, we know that $V(C) \cap K_i = \emptyset$, for $i = 0, 1$.

Assume there is a vertex $x \in K_0 \setminus K_1$. Then $(v_1, ..., v_{\ell-1}, x, v_\ell, ..., v_k, u_1)$ is a path in $G - K_1$, in contradiction to the fact that $K_1$ separates $u_1$ from $v_1$. So $K_0 \subseteq K_1$, and with the help of a symmetric argument we see that $K_0 = K_1$. In order to finish the proof it suffices to note that the paths $(v_1, ..., v_{\ell-1})$ and $(v_\ell, ..., v_k)$ ensure that there are components $A$ and $B$ as desired. $\qquad\square$

We now see that a graph that is obtained by gluing two graphs along a complete subgraph is chordal if and only if the two smaller graphs are.

**Lemma 22.** *Let $G$ be a graph, let $A, B, K$ be a partition of $V(G)$ such that $K$ is complete and there are no edges between $A$ and $B$. Then $G$ is chordal if and only if $G[A \cup K]$ and $G[K \cup B]$ are both chordal.*

*Proof.* As induced subgraphs of chordal graphs are chordal, we only need to show that if both $G[A \cup K]$ and $G[K \cup B]$ are both chordal, then also $G$ is. For this, it suffices to observe that any cycle of $G$ that contains vertices from both $A$ and $B$ has to pass twice through $K$. $\qquad\square$

We are now ready to prove Lemmas 18 and 19.

*Proof of Lemma 18.* Use Lemma 21 to see that the intersection $K$ of the neighborhoods of $u_0, v_0, u_1, v_1$ is either a clique or empty, and $G - K$ has two connected components $A, B$ such that $u_0, u_1 \in A$ and $v_0, v_1 \in B$ (after possibly changing the roles of $u_0$ and $v_0$).

As $G$ has no complete separation, and as there are no edges from $A$ to $B$, one of $A \cup K$, $B \cup K$ has to have a non-edge; because of symmetry we can assume this is $A \cup K$. According to Lemma 22, the subgraph of $G$ induced by $A \cup K$ is chordal. Then, according to Lemma 17, there is also a non-edge $uv$ with $u, v \in A \cup K$ having the additional property that $G(uv)$ is chordal. As $K$ is complete, we can assume that $u \in A$.

If there is no non-edge $uv$ as desired, we have that $G(u_i v_i)(uv)$ is non-chordal for some $i \in \{0,1\}$; by symmetry, let us assume $G(u_1 v_1)(uv)$ is non-chordal. So, we may apply Lemma 21 to see that the intersection $K'$ of the neighborhoods of $u, v, u_1, v_1$ is either a clique or empty, and $G - K'$ has two connected components $A', B'$ such that $u, u_1 \in A'$ and $v, v_1 \in B'$ (after possibly changing the roles of $u$ and $v$). Note that $K' \subseteq N(u_1) \cap N(v_1) \subseteq K$. Furthermore, $K \subseteq K'$, since $K'$ separates $u_1$ from $v_1$ and $K \subseteq N(u_1) \cap N(v_1)$. So $K = K'$.

In particular, $v \notin K$, that is, $v \in A$. So, as $v_1 \in B$, we know that $v, v_1$ lie in distinct components of $G - K$. However, we also have that $v, v_1$ belong to the same component (namely, $A$) of $G - K' = G - K$, a contradiction. So the desired non-edge $uv$ exists. $\qquad\square$

*Proof of Lemma 19.* We start by proving that $N(u_0) \cap N(v) = N(u_1) \cap N(v)$. For this assume there is an $i \in \{0,1\}$ and a vertex $x \in (N(u_{1-i}) \cap N(v)) \setminus N(u_i)$. Then $(u_{1-i}, x, v, u_i, u_{1-i})$ is an induced cycle of length 4 in $G(u_i v)$, a contradiction since this graph is chordal. This proves the equality, and we set $K := N(u_0) \cap N(v) = N(u_1) \cap N(v)$.

Because of Lemma 13, $K$ is complete and separates $u_0, u_1$ from $v$. Since $G$ does not have a complete separation, at least one of $G[A \cup K]$, $G[B \cup K]$ is not complete, but by Lemma 22 both are chordal. So by Lemma 17 and, again, Lemma 22, there is a non-edge $uw$ such that $G(uw)$ is chordal. Now, if $uw$ is not as desired, say because $G(u_0 v)(uw)$ is non-chordal, then there is an induced cycle $C$ of length at least 4 going through both $uw$ and $u_0 v$. However, $C$ has to meet $K$, which implies $C$ is a triangle, a contradiction. $\qquad\square$

## A.4  Proof of Theorem 4

**Symbolic method.**  To any sequence of numbers can be associated a generating function. For example, consider the sequence $(B_n)_{n \geq 0}$, where the Bell number $B_n$ denotes the number of partitions of size $n$. The exponential generating function of partitions is then defined as

$$P(z) = \sum_{n \geq 0} B_n \frac{z^n}{n!}.$$

The sum can be expressed at the partition level as well

$$P(z) = \sum_{p \in \mathrm{Partitions}(n)} \frac{z^{|p|}}{|p|!}.$$

The *symbolic method*, presented in [14], translates combinatorial descriptions into generating function equations. For example, since a partition is a set of nonempty sets, the exponential generating function of partitions is equal to

$$P(z) = e^{\exp(z)-1}.$$

The reader unfamiliar with the symbolic method can verify this result by working on recurrences at the coefficient level. In this example, choosing a partition of size $n$ is equivalent with choosing its number of blocks $k$, the size $n_j \geq 1$ of the $j$th block for each $1 \leq j \leq k$, and finally the content of those blocks, so

$$B_n = \sum_{k=0}^{n} \sum_{\substack{n_1 + \cdots + n_k = n \\ \forall j,\, n_j \geq 1}} \frac{1}{k!} \binom{n}{n_1, \ldots, n_k}.$$

Multiplying by $z^n/n!$, summing over $n$ and reorganizing the terms, we indeed recover

$$\sum_{n \geq 0} B_n \frac{z^n}{n!} = e^{\exp(z)-1}.$$

In the rest of the combinatorial proofs, the symbolic method will be preferred and we will let the motivated reader translate those proofs at the recurrence level.

```
def universal_ac(element_set):
    if is_empty(element_set):
        return EMPTY_PARTITION
    u = element_set.pop()
    block = {u}
    for v in element_set:
        if query(u, v):
            block.add(v)
            element_set.remove(v)
    partition = universal_ac(element_set)
    partition.add_block(block)
    return partition
```

Figure 2: The universal AC algorithm considers an element, compare it to the other elements to find its block, then partition the remaining elements. It is named after the graph theory convention to call "*universal*" a vertex linked to all vertices of a graph.

$q$-**analogs.**  Several families of integer identities have been generalized by introducing $q$-*analogs*. An introduction can be found in [12]. The $q$-analog of integer $n$ is defined as

$$[n]_q = 1 + q + \cdots + q^{n-1} = \frac{1 - q^n}{1 - q}.$$

The $q$-factorial of the integer $n$ is

$$[n]_q! = \prod_{j=1}^{n} [j]_q.$$

The $q$-exponential is defined as

$$e_q(z) = \sum_{n \geq 0} \frac{z^n}{[n]_q!}.$$

The $q$-Pochhammer symbol is defined as

$$(a; q)_n = \prod_{k=0}^{n-1} (1 - aq^k)$$

Observe that the $q$-analog reduce to their classic counterparts for $q = 1$.

**Lemma 23.** *The $q$-analogs satisfy the following classic identities*

$$[n]_q! = \frac{(q; q)_n}{(1 - q)^n}, \qquad \frac{1}{(x; q)_\infty} = \sum_{n \geq 0} \frac{x^n}{(q; q)_n}, \qquad e_q(x) = ((1 - q)x; q)_\infty^{-1}.$$

**Characterizing the complexity generating function.**  According to Theorem 3, all chordal AC algorithms share the same distribution on the number of queries for random partitions of size $n$ chosen uniformly. Thus, we study one particular chordal algorithm: the *universal AC algorithm*, presented in Figure 2. Let $\mathrm{Partitions}(n)$ denote the set of all partitions of size $n$, $|p|$ the size of the partition $p$, and $\mathrm{queries}(p)$ the number of queries used by the universal AC algorithm to reconstruct the partition $p$.

**Theorem 24.** *The generating function*

$$P(z, q) = \sum_{p \in \mathrm{Partitions}(n)} q^{\mathrm{queries}(p)} \frac{z^{|p|}}{|p|!}$$

*is characterized by the differential equation*

$$\partial_z P(z, q) = P(qz, q) e^{qz}$$

*and the initial condition $P(0, q) = 1$.*

*Proof.* Consider a partition $p$ of size $n$. Let $b$ denote the set of all elements of the block of $p$ containing $n$, except $n$. Let $r$ denote the partition $p$ without the block containing $n$. Then $p$ can be recovered from the pair $(r, b)$ as follows. The size $n$ of $p$ is $|r| + |b| + 1$, so the block containing $n$ in $p$ was $b \cup \{n\}$ and adding this block to $r$ recovers $p$. We have just proven that this construction is a bijection between the partitions and the relabeled pairs containing a partition and a set. This bijection translates into the following identity on the generating function $P(z)$ of partitions

$$\partial_z P(z) = P(z)e^z.$$

This is no surprise, as we already know the expression of this generating function

$$P(z) = e^{\exp(z)-1}$$

from the paragraph on the symbolic method, and it indeed satisfies this differential equation. However, the same approach is useful to study the generating function of the number of queries used by the universal AC algorithm.

Consider a partition $p$ and its decomposition as a pair $(r, b)$ described above. The universal AC algorithm starts by comparing one element, which is assumed to have the largest label without loss of generality, to the other elements. This requires $|r| + |b|$ queries. Then the algorithm is called recursively on the partition $r$. Thus, the generating function of partitions with an additional parameter $q$ marking the number of queries used by the universal AC algorithm is characterized by the differential equation

$$\partial_z P(z, q) = P(qz, q)e^{qz}.$$

If the initial partition is empty, then there are no queries to ask, which implies the initial condition $P(0, q) = 1$. $\qquad\square$

**Exact expressions.** The following theorem provides a solution for the differential equation from last Theorem.

**Theorem 25.** *Let* $\mathrm{Poch}(z, a, q)$ *denote the exponential generating function associated to the q-Pochhammer symbol*

$$\mathrm{Poch}(z, a, q) = \sum_{k \geq 0} (a; q)_k \frac{z^k}{k!},$$

*then the generating function of the universal AC algorithm complexity is*

$$P(z, q) = \mathrm{Poch}\left(-\frac{q}{1-q}z, \frac{1-q}{q}, q\right) e^{\frac{q}{1-q}z}$$

*Proof.* The function $f(z, q) = e^{\frac{q}{1-q}z}$ satisfies a similar differential equation

$$\partial_z f(z, q) = \frac{q}{1-q} f(qz, q)e^{qz}.$$

Thus, we investigate solutions of the differential equation from Theorem 24 of the form $P(z, q) = A(z, q)e^{\frac{q}{1-q}z}$. The differential equation on $P(z, q)$ implies the following differential equation for $A(z, q)$

$$\partial_z A(z, q) + \frac{q}{1-q} A(z, q) = A(qz, q).$$

with initial condition $A(0, q) = 1$. Decomposing $A(z, q)$ as a series in $z$

$$A(z, q) = \sum_{k \geq 0} a_k(q) \frac{z^k}{k!},$$

we obtain a recurrence on the $a_k(q)$

$$a_{k+1}(q) = -\frac{q}{1-q} a_k(q) + q^k a_k(q) = -\frac{q}{1-q}\left(1 - (1-q)q^{k-1}\right) a_k(q),$$

with $a_0(q) = 1$. We deduce

$$a_k(q) = \left(-\frac{q}{1-q}\right)^k \prod_{j=0}^{k-1}\left(1 - \frac{1-q}{q}q^j\right) = \left(-\frac{q}{1-q}\right)^k \left(\frac{1-q}{q}; q\right)_k.$$

To conclude, we observe that $\text{Poch}\left(-\frac{q}{1-q}z, \frac{1-q}{q}, q\right) e^{\frac{q}{1-q}z}$ is indeed solution of the differential equation characterizing $P(z, q)$. $\qquad\square$

Extracting the coefficient $n![z^n]$ from the solution, we obtain a first exact expression for $P_n(q)$ in the following Theorem, and another one will be provided in Theorem 27. This first expression is well suited for exact computations using a computer algebra system (we used [22] to verify our calculations). In particular, the $k$th factorial moment of the random variable $X_n$ counting the number of queries used by the universal AC algorithm on partitions of size $n$ chosen uniformly at random is

$$\mathbb{E}(X_n(X_n - 1)\cdots(X_n - k + 1)) = \frac{1}{B_n}\partial_{q=1}^k P_n(q).$$

**Theorem 26.** *The generating function of the number of queries used by the universal AC algorithm on partitions of size $n$ is*

$$P_n(q) = \left(\frac{q}{1-q}\right)^n \sum_{k\geq 0}\binom{n}{k}(-1)^k \left(\frac{1-q}{q}; q\right)_k.$$

We provide a second expression for the complexity generating function, more elegant and better suited for asymptotics analysis. It is a $q$-analog of the following classic formula for the Bell numbers, which counts the number of partitions of size $n$

$$B_n = \frac{1}{e}\sum_{m\geq 0}\frac{m^n}{m!}.$$

This formula is obtained from the generating function of partitions $P(z) = e^{\exp(z)-1}$

$$B_n = n![z^n]e^{\exp(z)-1} = \frac{n!}{e}[z^n]e^{\exp(z)} = \frac{n!}{e}[z^n]\sum_{m\geq 0}\frac{e^{mz}}{m!} = \frac{1}{e}\sum_{m\geq 0}\frac{m^n}{m!}.$$

**Theorem 27.** *The generating function of the number of queries used by the universal AC algorithm on partitions of size $n$ is*

$$P_n(q) = \frac{1}{e_q(1/q)}\sum_{m\geq 0}\frac{[m]_q^n}{[m]_q!}q^{n-m}$$

*The sum converges for $q > 1/2$.*

*Proof.* The $q$-Pochhammer generating function is rewritten as

$$\text{Poch}(z, a, q) = \sum_{k\geq 0}(a; q)_k \frac{z^k}{k!}$$

$$= (a; q)_\infty \sum_{k\geq 0}\frac{1}{(aq^k; q)_\infty}\frac{z^k}{k!}$$

$$= (a; q)_\infty \sum_{k\geq 0}\sum_{m\geq 0}[x^m]\frac{1}{(ax; q)_\infty}q^{mk}\frac{z^k}{k!}$$

$$= (a; q)_\infty \sum_{m\geq 0}a^m[x^m]\frac{1}{(x; q)_\infty}e^{q^m z}$$

Applying Lemma 23 we conclude

$$\text{Poch}(z, a, q) = (a; q)_\infty \sum_{m\geq 0}\frac{a^m}{(q; q)_m}e^{q^m z}$$

Injecting this in the expression of $P_n(q)$, we obtain

$$P_n(q) = n![z^n] \operatorname{Poch}\left(-\frac{q}{1-q}z, \frac{1-q}{q}, q\right) e^{\frac{q}{1-q}z}$$

$$= \left(\frac{q}{1-q}\right)^n \left(\frac{1-q}{q}; q\right)_\infty n![z^n] \sum_{m\geq 0} \frac{\left(\frac{1-q}{q}\right)^m}{(q;q)_m} e^{-q^m z} e^z$$

$$= \left(\frac{q}{1-q}\right)^n \left(\frac{1-q}{q}; q\right)_\infty \sum_{m\geq 0} \frac{\left(\frac{1-q}{q}\right)^m}{(q;q)_m} (1-q^m)^n$$

$$= q^n \left(\frac{1-q}{q}; q\right)_\infty \sum_{m\geq 0} \frac{\left(\frac{1-q}{q}\right)^m}{(q;q)_m} [m]_q^n.$$

To conclude, Lemma 23 is applied

$$P_n(q) = \frac{1}{e_q(1/q)} \sum_{m\geq 0} \frac{[m]_q^n}{[m]_q!} q^{n-m}.$$

We apply d'Alembert's criteria to find the values of $q$ for which this formal sum converges:

$$\lim_{m\to\infty} \frac{\frac{[m+1]_q^n}{[m+1]_q!} q^{n-m-1}}{\frac{[m]_q^n}{[m]_q!} q^{n-m}} = \frac{1-q}{q}$$

is smaller than 1 when $q > 1/2$. □

**Limit law.**

**Lemma 28.** *As $n$ and $m$ tend to infinity, $q = e^s$, $ms$ and $ns$ tend to $0$, we have*

$$[m]_q^n = m^n \exp\left(\frac{1}{2}nms + \frac{1}{12}nm^2 \frac{s^2}{2}\right) \left(1 + \mathcal{O}(nm^3 s^3 + ns)\right)$$

$$[m]_q! = m^m e^{-m} \sqrt{2\pi[m]_q} \exp\left(\frac{1}{4}m^2 s + \frac{1}{36}m^3 \frac{s^2}{2}\right) \left(1 + \mathcal{O}(m^4 s^3 + ms) + o(1)\right).$$

*Proof.* Let $S(x)$ denote the function $(e^x - 1 - x)/x$, then

$$[m]_q^n = \left(\frac{1-e^{sm}}{1-e^s}\right)^n = m^n \left(\frac{1+S(sm)}{1+S(s)}\right)^n = m^n e^{n\log(1+S(sm))-n\log(1+S(s))}.$$

We use the development

$$\log(1+S(x)) = \log\left(1 + \frac{x}{2} + \frac{x^2}{6} + \mathcal{O}(x^3)\right) = \frac{x}{2} + \frac{x^2}{24} + \mathcal{O}(x^3)$$

to obtain

$$[m]_q^n = m^n \exp\left(\frac{1}{2}nms + \frac{1}{12}nm^2 \frac{s^2}{2} + \mathcal{O}(nm^3 s^3 + ns)\right)$$

According to Moak [20], we have the following $q$-analog of Stirling formula when $x \to \infty$ while $x \log(q) \to 0$

$$\log(\Gamma_q(x)) = (x - 1/2)\log([x]_q) + \frac{\operatorname{Li}_2(1-q^x)}{\log(q)} + \frac{1}{2}\log(2\pi) + o(1)$$

where $\operatorname{Li}_2(z)$ denotes the Dilogarithm function

$$\operatorname{Li}_2(z) = \sum_{k\geq 1} \frac{z^k}{k^2}$$

We deduce

$$[m]_q! = [m]_q \Gamma_q(m) = [m]_q \exp\left((m - 1/2)\log([m]_q) + \frac{\text{Li}_2(1 - q^m)}{\log(q)} + \frac{1}{2}\log(2\pi) + o(1)\right)$$

$$= [m]_q^m \exp\left(\frac{\text{Li}_2(1 - q^m)}{\log(q)}\right)\sqrt{2\pi[m]_q}(1 + o(1))$$

The first part of the lemma provides

$$[m]_q^m = m^m \exp\left(\frac{1}{2}m^2 s + \frac{1}{12}m^3\frac{s^2}{2}\right)\left(1 + \mathcal{O}(m^4 s^3 + ms)\right).$$

The Dilogarithm is expanded as

$$\frac{\text{Li}_2(1 - q^m)}{\log(q)} = \frac{1}{s}\sum_{k \geq 1}\frac{1}{k^2}\left(-m\, s\, (1 + S(m\, s))\right)^k = -m\left(1 + \frac{1}{4}ms + \frac{1}{18}m^2\frac{s^2}{2} + \mathcal{O}(m\, s)^3\right).$$

Injecting those past two expansions in the previous one concludes the proof. $\qquad\square$

**Theorem 29.** *The asymptotic mean $E_n$ and standard deviation $\sigma_n$ of the number $X_n$ of queries used by the universal AC algorithm on a partition of size $n$ chosen uniformly at random are*

$$E_n = \frac{1}{4}(2\zeta - 1)e^{2\zeta} \quad and \quad \sigma_n = \frac{1}{3}\sqrt{\frac{3\zeta^2 - 4\zeta + 2}{\zeta + 1}e^{3\zeta}},$$

*where $\zeta$ is the unique positive solution of*

$$\zeta e^\zeta = n.$$

*The normalized random variable*

$$X_n^\star = \frac{X_n - E_n}{\sigma_n}$$

*follows in the limit a normalized Gaussian law.*

Those results are tested numerically in Figures 3 and 4.

*Proof.* To prove the limit law, we show that the Laplace transform $\mathbb{E}(e^{sX_n^\star})$ converges pointwise to the Laplace transform of the normalized Gaussian $e^{s^2/2}$. We have

$$\mathbb{E}(e^{sX_n^\star}) = e^{-sE_n/\sigma_n}\mathbb{E}(e^{sX_n/\sigma_n}) = \frac{e^{-sE_n/\sigma_n}}{B_n}P_n(e^{s/\sigma_n})$$

For any fixed real value $s$, we compute the asymptotics of $P_n(e^{s/\sigma_n})$. Let $q := e^{s/\sigma_n}$, so $q$ tends to 1, then

$$P_n(e^{s/\sigma_n}) = \frac{1}{e_q(e^{-s/\sigma_n})}\sum_{m \geq 0}\frac{[m]_q^n}{[m]_q!}e^{(n-m)s/\sigma_n}.$$

Motivated by the asymptotics from Lemma 28, we rewrite this expression as

$$P_n(e^{s/\sigma_n}) = \frac{1}{e_q(e^{-s/\sigma_n})}\sum_{m \geq 0}A_{n,s}(m)e^{-\phi_{n,s}(m)} \tag{2}$$

where

$$A_{n,s}(m) = \frac{[m]_q^n}{m^n \exp\left(\frac{1}{2}nm\frac{s}{\sigma_n} + \frac{1}{12}nm^2\frac{(s/\sigma_n)^2}{2}\right)}\frac{m^m e^{-m}\exp\left(\frac{1}{4}m^2\frac{s}{\sigma_n} + \frac{1}{36}m^3\frac{(s/\sigma_n)^2}{2}\right)}{[m]_q!}e^{(n-m)s/\sigma_n},$$

$$\phi_{n,s}(m) = -n\log(m) + m\log(m) - m - \frac{1}{4}(2n - m)m\frac{s}{\sigma_n} - \frac{1}{36}(3n - m)m^2\frac{(s/\sigma_n)^2}{2}.$$

The dominant contribution to the sum comes from integers $m$ close to the minimum of $\phi_{n,s}(m)$, so we study this function. The successive derivatives of $\phi_{n,s}(m)$ are

$$\phi'_{n,s}(m) = -\frac{n}{m} + \log(m) - \frac{1}{2}(n-m)\frac{s}{\sigma_n} - \frac{1}{12}(2n-m)m\frac{(s/\sigma_n)^2}{2},$$

$$\phi''_{n,s}(m) = \frac{n}{m^2} + \frac{1}{m} + \frac{s}{2\sigma_n} - \frac{1}{6}(n-m)\frac{(s/\sigma_n)^2}{2}$$

$$\phi'''_{n,s}(m) = -\frac{2n}{m^3} - \frac{1}{m^2} + \frac{(s/\sigma_n)^2}{12}$$

When $n$ is large enough, the second derivative of $\phi_{n,s}(m)$ is strictly positive for all $m > 0$, so the function is convexe. It reaches its unique minimum at a value denoted by $m(s)$ and characterized by $\phi'_{n,s}(m(s)) = 0$. Injecting the Taylor expansion

$$m(s) = m_0 + m_1\frac{s}{\sigma_n} + m_2\frac{(s/\sigma_n)^2}{2} + \cdots$$

in this equation, rewriting $n$ as $\zeta e^\zeta$ and extracting the coefficients of the powers of $s$, we obtain

$$m_0 = e^\zeta, \qquad m_1 = \frac{1}{2}\frac{\zeta-1}{\zeta+1}e^{2\zeta}, \qquad m_2 = \frac{1}{3}\frac{2\zeta^3 - 3\zeta^2 + 2}{(\zeta+1)^3}e^{3\zeta}$$

The dominant contribution to the sum defining $P_n(e^{s/\sigma_n})$ comes from values $m$ close to $m(s)$. The *central part* $C_n$ is defined as the integers $m$ such that $|m - m(s)| < c_n$. A heuristic proposed by [14] is to find $c_n$ such that

$$|\phi''_{n,s}(m(s))|c_n^2 \to +\infty \quad \text{and} \quad |\phi'''_{n,s}(m(s))|c_n^3 \to 0.$$

As $n$, and thus $\zeta$, tend to infinity, we have

$$m(s) \sim e^\zeta \sim \frac{n}{\log(n)}, \qquad |\phi''_{n,s}(m(s))| \sim \zeta e^{-\zeta} \sim \frac{\log(n)^2}{n}, \qquad |\phi'''_{n,s}(m(s))| \sim \zeta e^{-2\zeta} \sim \frac{\log(n)^3}{n^2},$$

so we choose $c_n = e^{3\zeta/5}$. Uniformly for $m$ in $C_n$, we have

$$m(s) = e^\zeta + \frac{1}{2}\frac{\zeta-1}{\zeta+1}e^{2\zeta}\frac{s}{\sigma_n} + \frac{1}{3}\frac{2\zeta^3 - 3\zeta^2 + 2}{(\zeta+1)^3}e^{3\zeta}\frac{(s/\sigma_n)^2}{2} + \mathcal{O}(e^{-\zeta/2}),$$

$$\phi_{n,s}(m) = -(\zeta^2 - \zeta + 1)e^\zeta - E_n\frac{s}{\sigma_n} - \frac{s^2}{2} + (\zeta+1)e^{-\zeta}\frac{(m-m(s))^2}{2} + \mathcal{O}(e^{-\zeta/4}),$$

$$A_{n,s}(m) = \frac{1}{\sqrt{2\pi e^\zeta}}\left(1 + o(1)\right).$$

In fact, computing the Taylor expansion of $\phi_{n,s}(m)$ at $m(s)$ came first. We chose the values of $E_n$ and $\sigma_n$ so that the coefficients in $s$ and $s^2$ are the ones presented in the above equation. The error term is then obtained using the Lagrange form of the remainder in Taylor's Theorem. We deduce the following asymptotics for the central part of the sum

$$\sum_{m \in C_n} A_{n,s}(m)e^{-\phi_{n,s}(m)} \sim \frac{1}{\sqrt{2\pi e^\zeta}}e^{(\zeta^2 - \zeta+1)e^\zeta + E_n\frac{s}{\sigma_n} + \frac{s^2}{2}}\sum_{m \in C_n} e^{-(\zeta+1)e^{-\zeta}\frac{(m-m(s))^2}{2}}.$$

Applying the Euler-Maclaurin formula to turn the sum into an integral, we obtain

$$\sum_{m \in C_n} A_{n,s}(m)e^{-\phi_{n,s}(m)} \sim \frac{1}{\sqrt{2\pi e^\zeta}}e^{(\zeta^2 - \zeta+1)e^\zeta + E_n s/\sigma_n + s^2/2}\int_{-c_n}^{c_n} e^{-(\zeta+1)e^{-\zeta}\frac{x^2}{2}}dx.$$

After the variable change $y = \sqrt{(\zeta+1)e^{-\zeta}}x$, observing that $\sqrt{(\zeta+1)e^{-\zeta}}c_n$ tends to infinity, the integral is approximated as a Gaussian integral and we conclude

$$\sum_{m \in C_n} A_{n,s}(m)e^{-\phi_{n,s}(m)} \sim \frac{1}{\sqrt{\zeta+1}}e^{(\zeta^2 - \zeta+1)e^\zeta + E_n s/\sigma_n + s^2/2}.$$

When we compare the asymptotics of the central part to the asymptotics of the Bell numbers (see, e.g., [14])

$$B_n \sim \frac{e^{(\zeta^2 - \zeta + 1)e^\zeta - 1}}{\sqrt{\zeta + 1}},$$

we see from Equation (2) that, as expected,

$$\frac{e^{-sE_n/\sigma_n}}{B_n} \frac{1}{e_q(e^{-s/\sigma_n})} \sum_{m \in C_n} A_{n,s}(m) e^{-\phi_{n,s}(m)} \sim e^{s^2/2}.$$

Let us now prove that the part of the sum corresponding to $m \leq m(s) - c_n$ is negligible compared to the central part. According to Lemma 28, we have

$$A_{n,s}(m) = \left(1 + \mathcal{O}(nm^3/\sigma_n^3)\right)\left(1 + \mathcal{O}(m^4/\sigma_n^3)\right) \frac{1}{\sqrt{2\pi[m]_q}} = \mathcal{O}(m^8).$$

Since $\phi_{n,s}(m)$ is convexe (for large enough $n$), we have $\phi_{n,s}(m) \geq \phi_{n,s}(m(s) - c_n)$ for all $m < m(s) - c_n$. Since

$$\phi_{n,s}(m(s) - c_n) = -(\zeta^2 - \zeta + 1)e^\zeta - E_n \frac{s}{\sigma_n} - \frac{s^2}{2} + (\zeta + 1)e^{-\zeta} \frac{c_n^2}{2} + \mathcal{O}(e^{-\zeta/4})$$

and $e^{-\zeta}c_n^2$ tends to infinity as $e^{\zeta/5}$, we obtain for all $m < m(s) - c_n$

$$\phi_{n,s}(m) \geq -(\zeta^2 - \zeta + 1)e^\zeta - E_n \frac{s}{\sigma_n} - \frac{s^2}{2} + \Theta(e^{\zeta/5}).$$

We conclude

$$\sum_{m < m(s) - c_n} A_{n,s}(m) e^{-\phi_{n,s}(m)} \leq \sum_{m < m(s) - c_n} \mathcal{O}(m^8) e^{(\zeta^2 - \zeta + 1)e^\zeta + E_n s/\sigma_n + s^2/2 - \Theta(\exp(\zeta/5))}$$

$$\leq e^{(\zeta^2 - \zeta + 1)e^\zeta + E_n s/\sigma_n + s^2/2} m(s)^9 e^{-\Theta(\exp(\zeta/5))}.$$

Since $m(s) \sim e^\zeta$, this result is exponentially small, with respect to $n$, compared to the central part. Let us now prove that the part of the sum beyond the central part is negligible as well. There is a constant $C$ such that for $n$ large enough and any $m \geq Ce^{3\zeta/2}$, we have

$$-\frac{1}{4}(2n - m)m \frac{s}{\sigma_n} - \frac{1}{36}(3n - m)m^2 \frac{(s/\sigma_n)^2}{2} \geq 0.$$

In that case, we obtain the simple bound

$$\phi_{n,s}(m) \geq -n\log(m) + m\log(m) - m \geq m - n\log(m).$$

Injecting this bound and $A_{n,s}(m) = \mathcal{O}(m^8)$ in the sum, we obtain

$$\sum_{m \geq Ce^{3\zeta/2}} A_{n,s}(m) e^{-\phi_{n,s}(m)} \leq \mathcal{O}(1) \sum_{m \geq Ce^{3\zeta/2}} m^{n+8} e^{-m}.$$

The sum is bounded by an integral and $n + 8$ integration by part are applied

$$\sum_{m \geq Ce^{3\zeta/2}} A_{n,s}(m) e^{-\phi_{n,s}(m)} \leq \mathcal{O}(1)(n+8)!(Ce^{3\zeta/2})^{n+8} e^{-C\exp(3\zeta/2)}$$

$$\leq \mathcal{O}(1)n^n (Ce^{3\zeta/2})^{n+8} e^{-C\exp(3\zeta/2)}$$

Since $n = \zeta e^\zeta$, we have $n^n = e^{\mathcal{O}(\zeta^2)\exp(\zeta)}$, so

$$\sum_{m \geq Ce^{3\zeta/2}} A_{n,s}(m) e^{-\phi_{n,s}(m)} \leq \mathcal{O}(1) e^{-\Theta(\exp(3\zeta/2))}$$

which is negligible compared to the central part of the sum. The last part we consider is $m(s) + c_n \leq m \leq Ce^{3\zeta/2}$. Since $\phi_{n,s}(m)$ is decreasing there and $A_{n,s}(m) = \mathcal{O}(m^8)$, we have

$$\sum_{m=m(s)+c_n}^{Ce^{3\zeta/2}} A_{n,s}(m) e^{-\phi_{n,s}(m)} = \mathcal{O}(e^{27\zeta/2}) e^{-\phi_{n,s}(m(s)+c_n)}$$

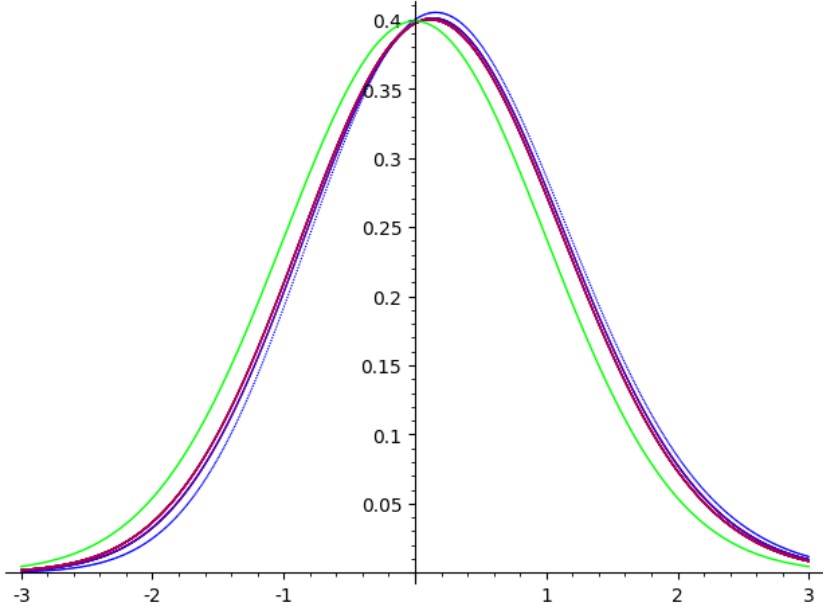

Figure 3: In green, the probability density function of the normal distribution. In blue, purple and red, the empirical density functions for the number of queries, normalized by their mean and standard deviation, for $n$ in $\{100, 300, 600\}$. We observe a slow convergence to the Gaussian limit law.

As for the case $m = m(s) - c_n$, we find

$$\phi_{n,s}(m(s) + c_n) = -(\zeta^2 - \zeta + 1)e^\zeta - E_n \frac{s}{\sigma_n} - \frac{s^2}{2} + \Theta(e^{\zeta/5})$$

and conclude

$$\sum_{m=m(s)+c_n}^{Ce^{3\zeta/2}} A_{n,s}(m)e^{-\phi_{n,s}(m)} = e^{(\zeta^2-\zeta+1)e^\zeta + E_n \frac{s}{\sigma_n} + \frac{s^2}{2}} \mathcal{O}(e^{27\zeta/2})e^{-\Theta(e^{\zeta/5})},$$

which is negligible compared to the central part. In conclusion, we have

$$\mathbb{E}(e^{sX_n^\star}) = \frac{e^{-sE_n/\sigma_n}}{B_n} P_n(e^{s/\sigma_n}) \sim \sum_{m\in C_n} A_{n,s}(m)e^{-\phi_{n,s}(m)} \sim e^{s^2/2}.$$

Since the Laplace transform of $X_n^\star$ converges pointwise to the Laplace transform of the normalized Gaussian law, $X_n^\star$ converges in distribution to this Gaussian. $\qquad\square$

## A.5  Proof of Theorem 6

Suppose first that $p_1 > \cdots > p_k$. At time $t$, we have identified classes $C_1^t, \ldots, C_k^t$, say with sizes $c + i^t = |C_i^t|$. It takes time at most $k\sqrt{n}$ to process the first $\sqrt{n}$ vertices, and at that point and all subsequent times we have (with exponentially small failure probability) that $c_i^t \sim p_i\sqrt{n}$ for each $i$, where $C_i^t$ consists of elements of type $i$. In particular $|C_1^t| > |C_2^t| > \cdots > |C_k^t|$. The expected number of comparisons used for each subsequent element is $\sum i = 1^k p_i$, as the probability that the element is of type $i$ is $p_i$, and in this case we need $i$ comparisons to place it in $C_i^t$.

The case where some of the $p_i$ are the same is similar, except that if $p_i = p_j$ then $c_i^t$ and $c_j^t$ may switch order over time. However this is fine: we need only have the property that if $p_i > p_j$ then $c_i^t > c_j^t$, and the same argument then works, possibly permuting colors for classes $i, j$ with $p_i = p_j$.

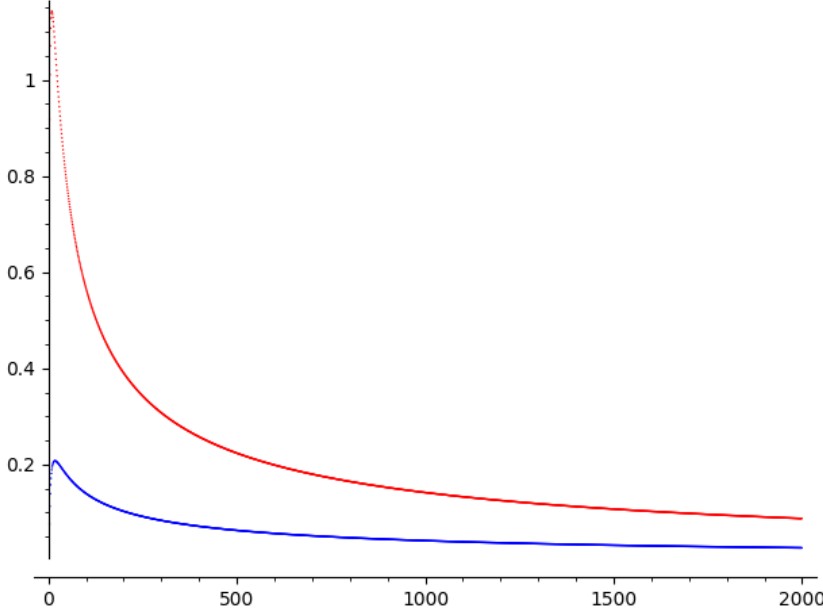

Figure 4: A plot testing the asymptotic mean and standard deviations stated by Theorem 4. Let $\alpha_n$ and $\beta_n$ denote the sequences defined by $\mathbb{E}(X_n) = \frac{1}{4}(2W(n) - 1 + \alpha_n)e^{2W(n)}$ and $\sigma(X_n) = \frac{1}{3}\sqrt{\frac{3W(n)^2 - 4W(n) + 2 - \beta_n}{W(n) + 1}}e^{3W(n)}$. Plot of $\alpha_n$ in blue, $\beta_n$ in red, for $n$ from 3 to 2000. We observe a slow convergence to 0.

### A.6  Proof of Theorem 12

The idea of the proof is as follows: we estimate the behaviour of the process until large linear time, and then note that it finishes in negligible time. Let us note first that in any partition of a set of size $t$, where $t$ is large, a fraction of at least (about) $1/k$ of the pairs lie inside classes. It follows that at each time step, we reduce the number of vertices by 1 with constant probability, and so the the algorithm finishes in expected time $O(kt)$. When handling an instance of size $n$, it is therefore enough to run the algorithm until $o(n)$ vertices remain, and then note that expected time to complete is still $o(n)$.

We consider an instance of size $n$, and analyze the following process. Begin with all vertices marked *active*. At each time step, pick (with replacement) a random pair $\{u, v\}$ and:

- If $u, v$ are active and from distinct classes $i, j$ then say we have generated an *ij-crossedge*.

- If $u, v$, are both active and in class $i$ then mark exactly one of $u$ and $v$ inactive.

- If one of $u, v$ is inactive, then do nothing.

Note that as the process runs, the number of active vertices is monotonic decreasing, and we are increasingly likely to choose pairs where one vertex is inactive. These contribute to the new process, but do not generate new comparisons between pairs. So we are looking at a (randomly) slowed down version of the random algorithm; but this makes the analysis much simpler!

Let $x_i(t)$ denote the number of active class $i$ vertices after $t$ time steps, and $x_{ij}(t)$ denote the number of $ij$-crossedges that are generated in the first $t$ time steps. Then at step $t + 1$, the probability that we pick two active vertices in class $i$ is

$$\binom{x_i(t)}{2} \Big/ \binom{n}{2} \sim \frac{x_i(t)^2}{n^2}.$$

Writing $p = p_i$, we estimate $x_i(t)$ via a function $x = x(t)$ satisfying the differential equation

$$x(0) = pn$$

$$x'(t) = -\frac{x(t)^2}{n^2}.$$

This has solution

$$x(t) = \frac{n^2}{t + n/p} = \frac{pn}{1 + pt/n}.$$

Note that at time $\lambda n$, as $\lambda$ gets large, we get

$$x(\lambda n) \sim \frac{pn}{1 + \lambda p}.$$

The actual value of $x_i(t)$ closely tracks the differential equation with very high probability. This follows straightforwardly from a standard application of the Rödl nibble or differential equation method (see for example [3]): we throw in the edges in batches of size $\epsilon n$, and note that a Chernoff bound implies that we stick close to the the differential equation as we are in total making $O(n)$ comparisons.

Now let's estimate the number of $ij$-crossedges. Using our estimates for $x_i(t)$ and $x_j(t)$, we see that the probability of an $ij$ crossedge at step $t + 1$ is

$$x_i(t)x_j(t) / \binom{n}{2} \sim \frac{2x_i(t)x_j(t)}{n^2} \approx \frac{2n^2}{(t + n/p_i)(t + n/p_j)}.$$

Let $p = p_i$ and $q = p_j$. We can model the growth of $x_{ij}(t)$ by a function $c = c(t)$ satisfying the differential equation

$$c(0) = 0$$

$$c'(t) = \frac{2n^2}{(t + n/p)(t + n/q)}.$$

If $p = q$, we have

$$c'(t) = \frac{2n^2}{(t + n/p)^2}$$

and

$$c(t) = \int_0^t c'(s)ds = \int_0^t \frac{2n^2}{(s + n/p)^2}ds = \left[\frac{-2n^2}{s + n/p}\right]_0^t,$$

giving

$$c(t) = 2pn\left(1 - \frac{1}{2 + 2pt/n}\right) \sim 2pn$$

as $t/n \to \infty$.

If $p \neq q$, we have

$$\begin{aligned}
c(t) &= \int_0^t \frac{2n^2}{(s + n/p)(s + n/q)}ds \\
&= \frac{2npq}{p - q}\int_0^t \frac{1}{s + n/p} - \frac{1}{s + n/q}ds \\
&= \frac{2npq}{p - q}\ln\left(\frac{1 + pt/n}{1 + qt/n}\right) \\
&\sim \frac{2npq\ln(p/q)}{p - q}
\end{aligned}$$

as $t/n \to \infty$. Combining this with the remarks at the beginning of the proof gives the result.

## A.7 Proofs of Section 2.3

Consider an AC algorithm applied to a set of size $n$. Let $Q$ denote the set of queries with their answers when the algorithm terminates. We associate to $Q$ a graph $G$ where each vertex corresponds to an element of the starting set, each edge is either *positive* or *negative* and corresponds to the answers from $Q$. As stated in Paragraph *Correcting errors*, a contradiction is detected in $Q$ if and only if $G$ contains a contradictory cycle (a cycle where all edges except one are positive). In that case, additional queries are submitted until the responsible answers are identified and corrected.

If $Q$ contains no more contradiction, it characterizes a set partition $p$ (otherwise, the AC algorithm has not terminated) of size $n$ with a number of blocks denoted by $b$. However, this partition might not be the correct solution if $Q$ contains undetected errors. Assume each answer is wrong with a small probability $p$ and let $\mathbb{P}(\text{undetected error} \mid Q)$ denote the probability that $Q$ contains errors but no contradictory cycle. Its Taylor coefficients at $p = 0$ are denoted by $(c_i)_{i \geq 0}$

$$\mathbb{P}(\text{undetected error} \mid Q) = c_0 + c_1 p + c_2 p^2 + \cdots$$

We would like $Q$ to contain few additional queries compared to the noiseless case, while ensuring that $\mathbb{P}(\text{undetected error} \mid Q)$ is small. Since $p$ is assumed to be small, the second conditions corresponds to minimizing the vector $(c_0, c_1, \ldots)$ for the lexicographic order. Indeed, if we consider a larger vector $(c_0', c_1', \ldots)$, then there is a positive $\epsilon$ such that for any $0 < p < \epsilon$, we have

$$c_0 + c_1 p + c_2 p^2 + \cdots < c_0' + c_1' p + c_2' p^2 + \cdots$$

Let $d_k$ denote the number of sets of $k$ edges from $G$ which could be switched (positive edges become negative ones, negative edges become positive ones) without creating a contradictory cycle. This corresponds to the number of ways to change $k$ answers in $Q$ and obtain a result without contradiction. Let $m$ denote the total number of edges in $G$. Then the probability that there is an undetected error is

$$\sum_{k \geq 0} d_k p^k (1-p)^{m-k}.$$

Since this probability is $c_0 + c_1 p + c_2 p^2 + \cdots$, we deduce for all $k \geq 0$

$$c_k = \sum_{j=0}^{k} d_j [x^k] x^j (1-x)^{m-j}.$$

The triangular shape of this system of equations implies that the vector $(c_0, c_1, \ldots)$ is minimal for the lexicographic order if and only if the vector $(d_0, d_1, \ldots)$ is minimal for the lexicographic order.

Since $G$ contains no contradictory cycle, we have $d_0 = 0$. If all positive components of $G$ not reduced to one vertex are 2-edge-connected (*i.e.* they have no vertex of degree 1), then switching a positive answer must create a contradictory cycle. If every pair of positive components from $G$ is linked by at least 2 negative edges, then switching a negative answer must create a contradictory cycle. Thus, if $G$ satisfies those two conditions, we have $d_1 = 0$.

In order to make $d_2$ vanish, we would need every positive component of $G$ to be 3-edge-connected, so to have minimal degree at least 3 (unless the component is reduced to one vertex). As we saw in Paragraph *Bounded number of errors* from Section 2.3, this would substantially increase the number of queries. Thus, we choose a different approach, fixing a parameter $r$ that influences the number of queries added compared to the noiseless case, then constructing $Q$ so that $d_2$ (and hence $c_2$) is minimized. We will first describe the structure of $G$ (and hence $Q$), then provide an algorithm reaching this structure.

First, we ensure that each pair of positive components of $G$ are linked by at least 3 negative answers. This adds at most $2\binom{b}{2}$ queries, but should in practice be small, as in our random models, in the noiseless case, all pairs of positive components are typically already linked by more than 3 negative edges. This ensures that there are no pairs of negative answers from $Q$ that can be switched to create a contradictory cycle. Thus, $d_2$ counts the number of pairs of positive edges that can be switched without creating a contradictory cycle. Let $n$ denote the number of vertices of $G$ and $m$ its number of positive edges. If the partition $p$ characterized by $G$ contains $b$ blocks, then the minimal possible value of $m$ is $n - b$. It corresponds to all positive components being trees and is reached in the noiseless case. Since we want $m$ to not grow too far from this lower bound, we assume $m < 3n/2$.

According to [6, Section 4], the structure of $G$ minimizing $d_2$ is a graph where all vertices have degree 2 or 3, and there is an integer $s$ such that the maximal paths or cycles containing only vertices of degree 2 all have length $s$ or $s - 1$. In practice, during the query of human experts, this constraint might be difficult to satisfy exactly. So let us denote by $r$ the average length of those 2-paths, and by $r'$ an upper-bound. The number of positive edges is then

$$m = n \left( 1 + \frac{1}{3r + 2} \right),$$

so the number of edges added compared to the noiseless case is

$$m - (n - b) = \frac{n}{3r + 2} + b.$$

The number of 2-paths is

$$\frac{3n}{3r + 2},$$

so a bound on $d_2$ is

$$\binom{r' + 1}{2} \frac{3n}{3r + 2}.$$

We now describe an algorithm reaching this desirable structure. It inputs a positive parameter $r$. The largest $r$ is, the smaller the number of additional edges is compared to the noiseless case, but also the higher the probability of undetected error becomes. We start with an AC algorithm designed for the noiseless case (the clique algorithm for example) and maintain the graph $G$ described above. In the first phase, the positive components of $G$ are trees. Whenever the noiseless algorithm proposes a query between two positive components of $G$, we choose one vertex on each of those components such that a positive answer would create a positive component that is a tree where all vertices have degree at most 3, and all 2-paths have length close to $r$. The second phase starts when the noiseless algorithm has terminated. Additional queries are added

- between positive components so that each pair is linked by at least 3 negative answers,
- between the leaves of each tree corresponding to a positive component of $G$. No vertex should be left with positive degree 1, the 2-paths should have length close to $r$, and each positive component should be 2-edge-connected.

If at any point a contradictory cycle is detected, queries cutting it in two are submitted until the conflict is resolved.