# OpenReview forum: "Active clustering for labeling training data"
_NeurIPS.cc/2021/Conference — NeurIPS 2021 Poster_

### Official Review · Reviewer_9dwD · 2021-07-12

**Rating:** 7
**Confidence:** 1

**Summary:**

The authors address the problem of pairwise clustering, where users are given pairs of instances and need to decide whether they belong to the same class or not. Given the human feedback, a clustering is computed. The authors show the importance of chordal algorithms for complexity and analyze random and clique algorithms.

**Limitations And Societal Impact:**

Yes.

**Main Review:**

The technical contribution is very strong and provides elaborate insights in active clustering mechanisms. The paper is overall very well written and, given the complexity, easy to understand, visualizations provide additional intuition.

Some practical issues, like, e.g., dealing with noisy queries/answers however remain unclear. Though I agree that space is short, I would be interested in learning more about the integration of the provided strategy.

**Time Spent Reviewing:**

3

---

### Official Review · Reviewer_onZc · 2021-07-15

**Rating:** 7
**Confidence:** 3

**Summary:**

Given an unknown partition of the data, either randomly drawn uniformly from the space of partition, or complying to a distribution of items in cluster, this paper investigates the problem of having an optimal querying strategy to recover the aforesaid partition.

The paper explores this from a theoretical perspective, showing that the chordal property is a fundamental requirement in the first case (uniform sampling of partition), and suggesting that it may also be very useful in the distribution case (which is much closer to actual supervised learning).

**Limitations And Societal Impact:**

Yes they have

**Main Review:**

The paper is very well-written, and although it is not exactly my field of expertise, I could easily follow it and get the main intuition /results, and have a reasonable check at the proofs (in the amount of time I could allocate to the paper), which seem correct to me.

Given the caveat related to my expertise, it is hard for me to really give an opinion about the originality in the paper, that is more about elicitation or recovering of partitions in an optimal sequence of queries than it is about supervised learning. I thin the results are really nice, but I cannot really say if some works in operations research or combinatorial optimisation have not considered similar problems before.

I have only two main comments about this paper, who are not directly related to the main results, but rather to the presentation and to practical considerations:

* The first is that the paper content seems to me completely disconnected from the initial motivation of the paper, which is to perform elicitation/active learning for supervised learning problem through comparative/pairwise queries (BTW, I am not convinced that these latter are always easier to solve for an expert). However, once the introduction is passed, there is nothing in the paper telling us how the provided (nice) results do connect to supervised learning, nor any evidence that they would do a better job at supervised learning than classical active learning techniques (as no experiments are present). I therefore do not perceive active learning in supervised setting as the main motivation of the paper, and would be actually surprised if it actually was, given the absence of experimental comparison or even theoretical insight as to how this would relate to, e.g., minimizing an expected loss function. In my opinion it would be more honest to mention supervised learning in the conclusions as a possible application of the presented results, rather than stating it as an initial motivation for the study, which rather appear artificial. I do think the results stand on their own, and I would avoid having a selling point that is not really followed by concrete application to it.

* The second is that while we do have nice theoretical results about various possible algorithms and their optimality, one could wonder which of the two proposed algorithms (that could have been given as algorithm in a proper floating environment, at least in the appendices) actually works better and under which conditions in practice? It is one nice thing to provide theoretical arguments in favour of different methods, but practical/numerical results may actually tells a different story. It would therefore be a nice complement to at least have some experimental results, even simulated.


**Time Spent Reviewing:**

6h

---

> ### Author Response · Authors · 2021-08-10
> **Motivation and which algorithm to choose**
>
> Thank you for your precise and positive review.
>
> The initial motivation of this work indeed came from an industrial application. My company wanted to automatically answer the most frequent tickets submitted by its clients (tickets describe problems met with our products). One of the feedbacks from the experts answering queries to classify the training data was that pairwise comparison was easier (on this particular application) than direct labeling. We were surprised by the lack of references on the underlying theoretical problem (active clustering), found scientific depth in it, and decided to tackle it.
>
> The clique algorithm is chordal, thus optimal in the uniform model, and conjectured optimal also for the fixed distribution model (although using it limits our ability to parallelize the queries between the experts, which is useful in practice). Estimating the number of queries boils down to finding the random model that best represents the training data (uniform or fixed distribution). One way to decide is to apply the following thought experiment. Imagine classifying half of the items first. When the second half is classified, how many new classes will be discovered? If we expect a small number, then the fixed distribution model is better suited to model our training data set than the uniform model.

---

### Official Review · Reviewer_sREn · 2021-07-16

**Rating:** 5
**Confidence:** 4

**Summary:**


The authors propose an active clustering algorithm where the oracle annotate examples in must-link, cannot-link fashion. Assuming that the cost of labeling pairwise relationships of unlabeled data is less costly than annotating examples.  They propose to represent the pairwise labeling process as an n-set partition overall the unlabeled set of size n. They study the distribution of the number of queries of the proposed with the objective to bound the average number of queries needed by an AC algorithm to recover the n-set partition, they suggest a first setting where the partition is assumed to be drawn from a uniform random distribution of all possible partitions over the n-set, they show that if the process of labeling forms a chordal graph for all queries in the AC algorithm, then the average complexity in terms of number of queries is minimal and comparable for all  AC algorithms with chordal graph constraint, the argument made to suggest that pairwise labeling is always better than sample labeling in this setting is unclear. For the second setting, they assume that the distribution of random partitions generates graphs with a fixed number of blocks which relates to the number of classes and class distribution of the unlabeled set, they show that the clique AC algorithm exhibits minimal linear complexity convergence in terms of number of query in this case. Lastly,  they propose a method for correcting errors in the process of labeling pairwise examples, they show that we can detect k labeling errors under strong connectivity conditions depending on k on the partition graph.


**Limitations And Societal Impact:**

Yes

**Main Review:**

The paper is well written, however there are no empirical results to back up the study
* Underlying assumption made in this work (all unlabeled examples are uniformly informative) is too strong. In most active learning scenarios the cost of labeling (sample labeling or pairwise labeling) all the unlabeled set is costly,  in this case it is interesting to learn predictors with a labeling cost << n .
A naive solution for AC algorithms would be to learn a classifier after every query t made, the stopping criterion will depend on the accuracy of the learned classifier.  Such solutions do not take into account the informativity of different examples to solve the classification task in hand.
* Connectivity constraints for labeling error detection are hard to control empirically for larger k.


**Time Spent Reviewing:**

4

---

> ### Author Response · Authors · 2021-08-10
> **Training a classifier to identify the best queries is indeed useful**
>
> We agree that additional information on the items, such as a classifier trained on the answers to the first queries, improves the active clustering algorithms. One could even use less than n queries to classify n items if the classifier was trusted on the answers it considers as most certain. However, there is an explosion in the number of variants: should we train a classifier or learn a distance? What model should we use for the informativity of the potential queries? How do we compute the confidence of the classifier? In this paper, we chose to investigate the underlying fundamental problem that all those variants rely on. We analyzed what could be done without additional information (here, learning the classifier or distance), to inform future studies on the variants. Already in this simple model, we identified two different regimes depending on the model that best represents the training data (uniform set partition or fixed distribution).

---

### Decision · Program_Chairs · 2021-09-27

**Decision:**

Accept (Poster)

**Comment:**

This is a well-written paper that makes a nice theoretical contribution in analyzing two models for the problem of gathering training data, where the human experts are able to answer pairwise queries.  There is a noted lack of experiments, but the theoretical contributions alone are enough to warrant acceptance.